# An implantable piezoelectric ultrasound stimulator (ImPULS) for deep brain activation

Jason F. Hou [1,7], Md Osman Goni Nayeem [1,7], Kian A. Caplan[2], Evan A. Ruesch[3], Albit Caban-Murillo[3], Ernesto Criado-Hidalgo [4], Sarah B. Ornellas [1], Brandon Williams[5], Ayeilla A. Pearce [2], Huseyin E. Dagdeviren[6], Michelle Surets [3], John A. White [5], Mikhail G. Shapiro [4], Fan Wang [2], Steve Ramirez [3] & Canan Dagdeviren [1] ✉

Precise neurostimulation can revolutionize therapies for neurological disorders. Electrode-based stimulation devices face challenges in achieving precise and consistent targeting due to the immune response and the limited penetration of electrical fields. Ultrasound can aid in energy propagation, but transcranial ultrasound stimulation in the deep brain has limited spatial resolution caused by bone and tissue scattering. Here, we report an implantable piezoelectric ultrasound stimulator (ImPULS) that generates an ultrasonic focal pressure of 100 kPa to modulate the activity of neurons. ImPULS is a fully-encapsulated, flexible piezoelectric micromachined ultrasound transducer that incorporates a biocompatible piezoceramic, potassium sodium niobate $[(K,Na)NbO_3]$. The absence of electrochemically active elements poses a new strategy for achieving long-term stability. We demonstrated that ImPULS can i) excite neurons in a mouse hippocampal slice ex vivo, ii) activate cells in the hippocampus of an anesthetized mouse to induce expression of activity-dependent gene c-Fos, and iii) stimulate dopaminergic neurons in the substantia nigra pars compacta to elicit time-locked modulation of nigrostriatal dopamine release. This work introduces a non-genetic ultrasound platform for spatially-localized neural stimulation and exploration of basic functions in the deep brain.

Precise and reversible spatiotemporal control of neural activity is the ultimate goal of neurostimulation strategies both for therapeutic applications and neuroscience research. Current neurostimulation strategies can be broadly divided into two categories: (i) non-invasive and (ii) invasive. Some existing non-invasive methods used in clinical treatment are transcranial magnetic stimulation (TMS)[1], transcranial current stimulation (TCS)[2], and transcranial-focused ultrasound (tFUS)[3]. While these methods can avoid surgery and associated recurrent risks[4], TMS and TCS encounter scattering of electromagnetic energy through bone and tissue attenuation[5,6]. Ultrasound is a

[1]Media Lab, Massachusetts Institute of Technology, Cambridge, MA 02139, USA. [2]Department of Brain and Cognitive Sciences, McGovern Institute for Brain Research, Massachusetts Institute of Technology, Cambridge, MA 02139, USA. [3]Department of Psychological and Brain Sciences, The Center for Systems Neuroscience, Boston University, Boston 02215 MA, USA. [4]Division of Chemistry and Chemical Engineering, California Institute of Technology, Pasadena, CA 91125, USA. [5]Center for Systems Neuroscience, Neurophotonics Center, Department of Biomedical Engineering, Boston University, 610 Commonwealth Ave., Boston, MA 02215, USA. [6]Department of Neurosurgery, Faculty of Medicine, Istanbul University, Istanbul 34093, Turkey. [7]These authors contributed equally: Jason F. Hou, Md Osman Goni Nayeem. ✉e-mail: canand@media.mit.edu

modality that has been used by the medical community for half of a century as a tissue-safe medium of energy transduction. Conformable ultrasound electronics interface intimately with soft tissues to image, deliver drugs to, or stimulate organs[7–9]. Unobstructed transcranial-focused ultrasound (tFUS) beams can achieve millimeter-scale resolution in neural tissue and penetrate several centimeters to excite neurons by affecting mechanoreceptive and other membrane-bound ion channels[10–13]. Therefore, the ability to safely evaluate potential stimulation targets, and with adjustable parameters such as frequency and acoustic intensity, make it an advantageous approach[14,15] for neurostimulation therapy in patients with conditions such as Alzheimer's disease, epilepsy, and depression. Ultrasound, when transmitted from outside the human skull, faces significant scattering and reflection from the skull's high acoustic impedance[16], which can cause off-target stimulation via conduction through bone and auditory pathways[17,18] and even traumatic, irreversible brain injury[19]. To achieve a balance between skull transmission and spatial selectivity, most significant modulations of neurons with ultrasound have been reported at frequencies less than 1 MHz[15,20], and particularly with 500 kHz with pressures at or above 100 kPa[14,21–23].

Implantable devices allow electrical and chemical modulation of the brain, leading to significant advancements in treating neurological and psychiatric disorders[24–27]. Electrical deep brain stimulation (DBS) can induce reversible activation of neurons but is limited by anisotropic charge transfer across the brain's ionic medium to regions proportional to the size of the electrode[28,29]. Both the charge provided by these electrodes and the sensitivity of surrounding tissue can decrease significantly over time due to biofouling and corrosion, which limits the longevity of the device[28,30,31]. Using various frequencies of light, an optogenetics approach provides minimally invasive neurostimulation with high spatiotemporal resolution and cell-type specificity[32]. Development of safe, widespread non-immunogenic delivery in the brain remains a challenge for clinical translation[33,34]. A miniaturized, non-genetic platform for localized neurostimulation is therefore needed to fill the gap for next-generation neural interfaces to reach high standards of safety and longevity. Recently, several reports of miniaturized ultrasonic neurostimulation devices have shown that directed ultrasound energy can activate cultured neurons[35] and neurons in brain slices[36]. However, the proposed platforms are not suitable for implantation in the deep brain due to their rigid form factors, material composition, or high-power requirements. A scalable implant system with no electrochemically active elements that has the capability to non-genetically and locally modulate neurons in deep subcortical brain regions is needed to fill the translational gap.

Here, we report an implantable piezoelectric ultrasound stimulator (ImPULS) that delivers acoustic energy directly and precisely to populations of neurons in deep brain regions. Our key findings include the design and development of a low-power, micron-scale flexible piezoelectric micromachined ultrasound transducer (30 μm thick with an outline width of 140 μm where the diameter of the active piezo element is 100 μm) that can evoke neurons adjacent to the transducer in the deep brain. The ImPULS (i) uses biocompatible piezoelectric thin film of potassium sodium niobate (KNN)[37] as an active element suspended over an air-filled cavity as an acoustic backing, (ii) generates ultrasound at a pressure of 59.2 kPa at 15 μm away from device (100 kPa adjacent to the transducer) for a single element, (iii) remains functional after 7 days in an accelerated (75 °C) phosphate-buffered saline (PBS) solution without incurring significant electrical and mechanical degradation, and (iv) does not cause temperature rise above safe tissue thresholds during ultrasound generation[10]. We demonstrate the stimulation of neurons in a coronal hippocampal slice ex vivo captured by two-photon microscopy and activation of hippocampal cells in anesthetized mice to induce expression of the activity-dependent gene cFos across acute and 14-day timescales.

Furthermore, in vivo stimulation of dopaminergic neurons in the substantia nigra pars compacta (SNc) with the ImPULS elicits time-locked modulation of striatal dopamine release, highlighting the ImPULS as a potent neuromodulatory tool.

## Results

### Design, fabrication, and characterization of the ImPULS

The implantable piezoelectric ultrasound stimulator, ImPULS, is a flexible piezoelectric micromachined ultrasound transducer (pMUT), that is surgically implanted into the brain. A schematic of the ImPULS implanted into a subcortical region of a wild-type mouse is shown in Fig. 1a. Upon application of an alternating voltage, the ImPULS generates an ultrasound beam and excites nearby neurons as described in detail in later sections. The ImPULS applies a transfer printing process that enables the fabrication of implantable ultrasound stimulators using biocompatible piezoceramics insulated with durable polymers. This involves wet etch patterning and release of target piezoelectric films from host Silicon (Si) wafers and integration onto polymer SU-8 by transfer printing (Supplementary Figs. 1 and 2). The substrates can be engineered to serve the requirements of the application such as chemical resistance, stiffness, and biostability. Figure 1b is the peeled view of the ImPULS revealing each constituent layer; comprising SU-8 as substrate (0.8 μm in thickness), encapsulation and backing layers (0.5 μm and 15 μm in thickness, respectively); piezoelectric KNN layer (1 μm in thickness, and 100 μm in diameter); chromium/gold (Cr/Au, 10/250 nm in thickness) and platinum (Pt, 100 nm in thickness) serving as top and bottom electrodes, respectively and Cr/Au as metal interconnects (10/250 nm in thickness). In order to maximize the vibration amplitude of the active membrane, the dimensions of the element geometries are designed to replicate a pMUT with a pinned boundary device structure[38]. Compared to bulk piezoceramics and silicon-based pMUTs, the ImPULS has a thinner profile and lower Young's Modulus which better couples to soft brain tissue. Without a pMUT, a simple thickness-mode resonance-dependence makes piezoceramic devices at the 500 kHz range millimeters thick and unfavorable for minimally-invasive neurostimulation. We chose to use a single-element pMUT for this study to target adjacent neuron somas in the 50 μm hemispherical radius of the transducer as will be presented in simulation later in Fig. 2d. However, the ImPULS fabrication process can be scaled to produce flexible arrays that target larger regions of the brain tissue. A microfabricated pMUT array of 9 elements, where each element has the same dimension as the ImPULS active piezo unit and the resonant frequency is in the 500 kHz range is shown in Supplementary Fig. 3. As the microfabrication of suspended free-standing membranes remains a challenge using standard processes, we fabricated the devices in an inverted manner, which included a final bonding of a backing layer and exposure through a thin transparent polyethylene terephthalate (PET) layer to seal the air-filled cavity. The cavity and backing layers are also designed to increase the stiffness of the ImPULS for acute implantation without an insertion shuttle (see "Methods" section for a full description of the fabrication process).

We choose KNN as the lead-free piezoelectric layer due to (i) its comparably high piezoelectric coefficients ($e_{31}$, $d_{31}$) and durability (DC stress lifetime: >24 h at 200 °C and 30 kV/cm, and Curie temperature of 350 °C) that exceeds commercially available doped lead zirconate titanate (PZT) and polymer-based piezoelectric, (ii) its proven biocompatibility and non-toxicity, and (iii) its commercial availability. The initial P-E characteristic of KNN before microprocessing is shown in Supplementary Fig. 4.

The microfabricated unit is connected to a printed circuit board (PCB) using an anisotropic conductive film (ACF) based cabling to complete the final device fabrication (Fig. 1c). A colorized scanning electron microscopy (SEM) image of the cross-section of the ImPULS shown in Fig. 1d depicts the air-filled cavity and encapsulation of the active electrical elements.

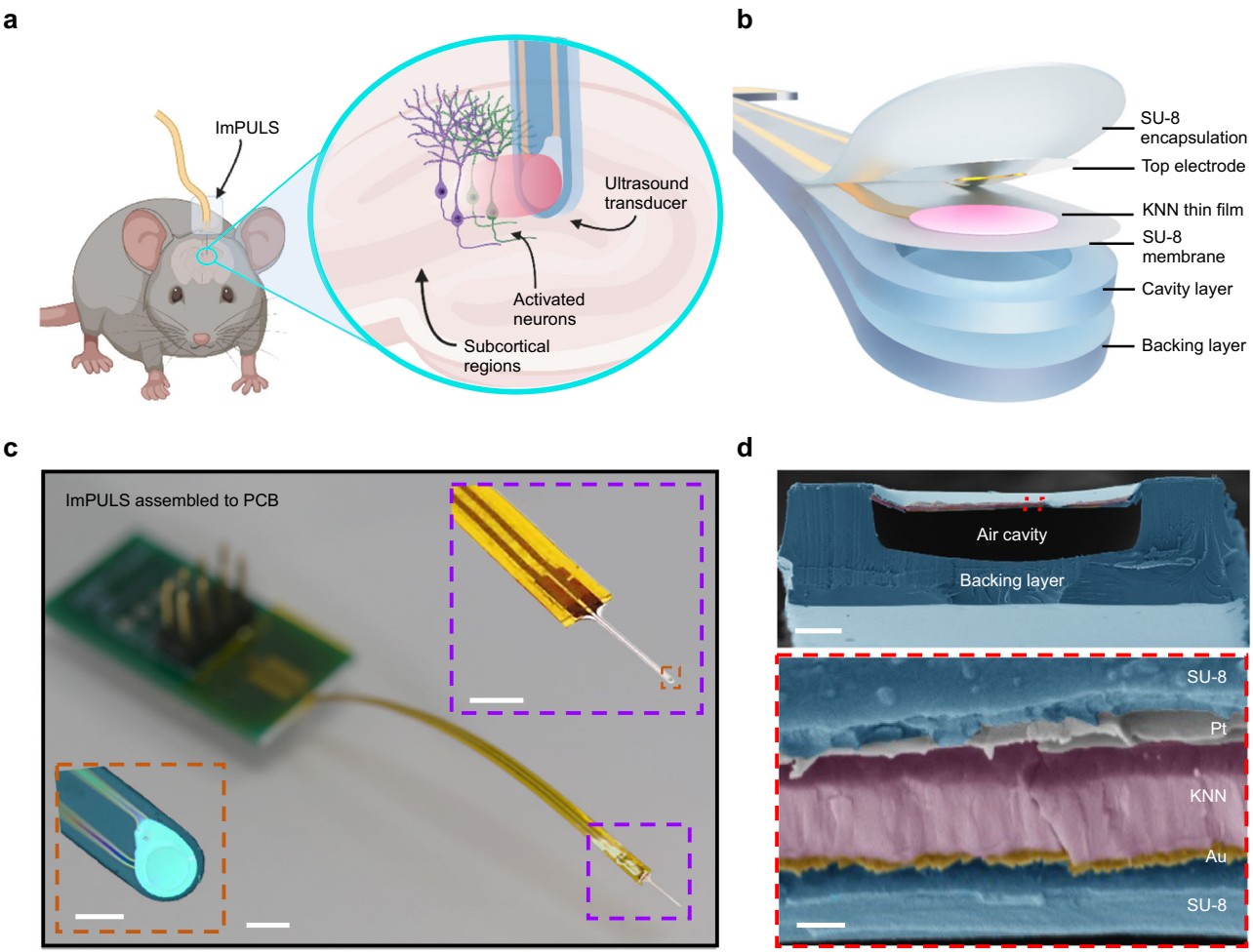

**Fig. 1 | An implantable piezoelectric ultrasound stimulator: ImPULS.**
**a** Schematic illustration of an implantable piezoelectric ultrasound stimulator (ImPULS) implanted in a subcortical brain region of a wild-type mouse. A magnified view showing the activated neurons with ultrasound application. Schematic created with BioRender.com, released under a Creative Commons Attribution-NonCommercial-NoDerivs 4.0 International license. **b** Schematic of a peeled view of the ImPULS, revealing each layer. The ImPULS is a piezoelectric micromachined ultrasound transducer (pMUT) structure where biocompatible potassium sodium niobate (KNN) is sandwiched between two thin SU-8 layers, and an air-filled cavity and a backing layer is formed underneath the piezoelectric thin-film membrane. **c** Optical image of the ImPULS assembled with flexible ACF cable and a custom printed circuit board (PCB) with a magnified view of the ImPULS probe (right, top inset) and further zoomed version of the tip of the probe (ultrasound unit) under a microscope (bottom, left inset). Scale bars, 5 mm, 2 mm, and 100 μm, respectively. **d** Colorized cross-sectional scanning electron microscope (SEM) image of the ImPULS. The device consists of (1) SU-8 encapsulation layer, (2) Top Pt electrode, (3) KNN thin film, (4) Bottom Au electrode, (5) SU-8 membrane layer, (6) Air-cavity, and (7) SU-8 backing layer. Scale bars, 20 μm and 500 nm, respectively.

To optimize the device parameters systematically, the electro-mechanical properties of the ImPULS were investigated before surgical implantation. Deionized water serves as a representative testing medium due to its similar acoustic properties to brain tissue[39] and its similar effects in terms of the resonant frequency shift of ImPULS, as we validated in 0.6% agar gel phantoms mimicking brain tissue. The electrical impedance and phase angle spectra of the ImPULS measured in air and water medium are shown in Fig. 2a. The resonance frequency of the ImPULS when submerged in water or implanted in phantom/mice brain has a pronounced but consistent shift due to the hydro-static forces exerted on the flexible material surrounding the entire device and the piezoelectric thin film itself[40,41]. Accordingly, we observe consistent decreases in resonance frequency of 40.4%, from 840 kHz to 500 kHz, in air versus water and agarose gel mediums, respectively. Similar resonance behavior of the ImPULS in air, water, and agarose gel mediums was observed under a scanning laser Doppler vibrometer (LDV) (Supplementary Fig. 5). We determined the resonant electromechanical behavior of ImPULS with a LDV by applying a periodic chirp excitation ranging from 100 kHz to 2 MHz to the device and measuring the frequency spectrum of the resulting

vibrations (Fig. 2b bottom). Once the resonance frequency was determined, a pure sinusoidal signal was applied to the device to determine the maximum displacement of the piezoelectric membrane. A displacement of 137 nm at a resonance frequency of 840 kHz (air medium) and displacement of 230 nm (water medium) was achieved upon application of a 4 V (p-p) sinusoidal signal.

The pressure output of a pMUT device is proportional to the center displacement of the vibration plate, and the center displacement represents the point of maximum mechanical energy transduction[38]. To optimize the device performance, we fabricated the ImPULS with different cavity sizes. As shown in Supplementary Fig. 6a, the resonance frequency decreases with the increase in cavity diameter. It has been reported that the most significant modulation of neurons occurs with the application of ultrasound at frequencies less than 1 MHz[15,20]. We, therefore, choose to characterize devices with a cavity size of 105 μm that has a resonance frequency of ~500 kHz. To investigate the effect of applied voltage on the transducer, we varied voltage from 2 V to 10 V (p-p) and recorded the peak displacement using a laser Doppler vibrometer (LDV) in both air and water mediums (Fig. 2c). The detailed experimental setup is described in

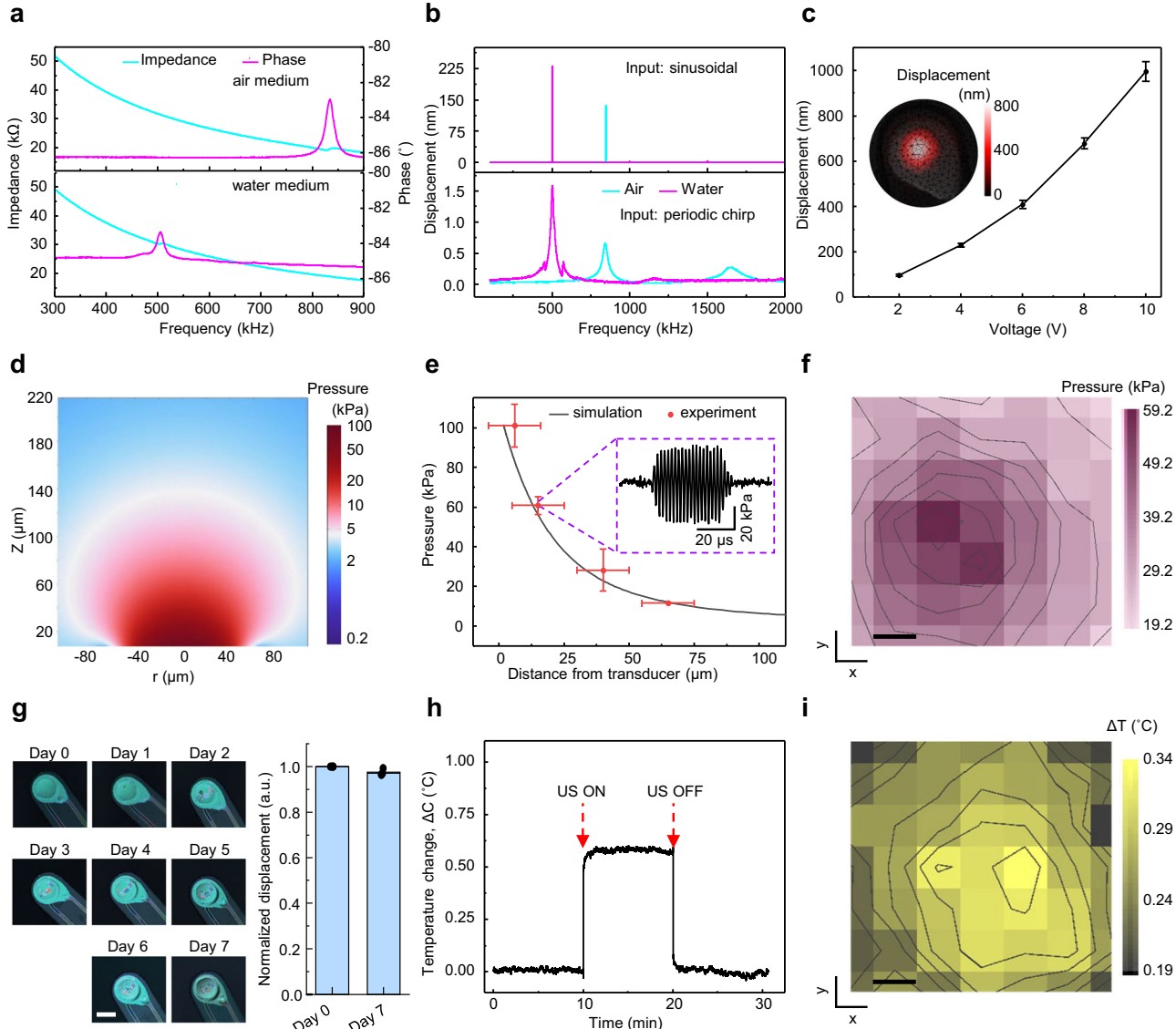

**Fig. 2 | Characterization of implantable piezoelectric ultrasound stimulator (ImPULS). a** The impedance and phase angle spectra of the ImPULS in air and water, showing the resonance frequency in both mediums. **b** Displacement of ImPULS in air and water mediums measured using laser Doppler vibrometer (LDV) at 4 V (p-p) when the inputs are a periodic chirp (bottom) and a sinusoidal signal (top). **c** Displacement of ImPULS as a function of input voltage (p-p) with inset showing two-dimensional (2-D) point scan of displacement indicating the lateral resolution of the beam of the device. Error bar represents the standard deviation in measurement, $N = 3$ devices. Data represents mean values ± SD. **d** Simulated acoustic pressure profile of the ImPULS showing a spherical pressure distribution. **e** Comparison of simulated and experimentally measured pressure using a fiber-optic hydrophone at different distances. Error bar represents the standard deviation in measurement, $N = 3$ devices. Data represents mean values ± SD in the x and y directions. **f** 2-D mapping of pressure generated by the ImPULS measured at $z = 15\,\mu m$. Scale bar, 25 μm. **g** Microscopic image of the ImPULS taken each 24 h apart during the aging test (left), and normalized displacement of ImPULS before the start of the test and after 7 days. Scale bar, 100 μm. Error bar represents a standard deviation in measurement, $N = 3$ devices. Data is normalized to the displacement measured on day 0 per device. Day 8 data represents the mean displacement ± SD. (a.u. arbitrary units). **h** Temperature change in water medium when a continuous sinusoidal signal of 500 kHz at 20 V (p-p) applied to the ImPULS. Ultrasound was 'off' for 10 min, 'on' for 10 min and 'off' for 10 min. **i** 2-D mapping of temperature generated by the ImPULS measured at $z = 15\,\mu m$. Scale bar, 25 μm.

Supplementary Methods and Supplementary Fig. 7. As shown in Fig. 2c, the output peak displacement increased from 98 nm to 995 nm when the applied voltage changed from 2 V to 10 V. A two-dimensional (2-D) point scan of the displacement was used to characterize the lateral resolution of the focal point, which reaches max intensity within an 80 μm diameter (Fig. 2c, inset). The movement and shape of membrane deformation can be visualized from a 3-D reconstruction of the 2-D scan as shown in Supplementary Fig. 8. Notably, the increase in cavity size leads to higher displacement as shown in Supplementary Fig. 6b, which can be attributed to the reduced piezoelectric diaphragm stiffness with the increased cavity size. Although pMUT

devices are commercially fabricated in Si substrates, our simulations indicate that it would be inefficient to create a KNN-based pMUT on a silicon substrate that is able to target the 500 kHz range without compromising its maximum width dimension and/or output pressure (Supplementary Note 2 and Supplementary Fig. 9).

To understand the generated pressure profile, we performed finite element analysis (FEA) using the COMSOL Multiphysics simulator (see "Methods" section and Supplementary Note 5 for parameters used in the simulation). As shown in Fig. 2d, the maximum pressure adjacent to the transducer can be as high as 100 kPa and decreases gradually following a spherical pressure distribution. We measured the pressure

generated by the ImPULS using a fiber-optic hydrophone positioned with its sensing element facing the device. ImPULS, with a diameter of 100 μm and 1 μm thick piezoelectric thin film, creates a near-field pressure within 10 μm and a far-field region around 100 μm in length. Measurements with an optically-coupled sensor for pressure is imperative because electromagnetic interference (EMI) generated by the wires and piezoelectric at distances less than 100 μm degrades the sensitivity of conventional piezoceramic hydrophones and over-powers the signal of interest. Therefore, we utilized a fiber-optic hydrophone mounted on a 3-axis stage that can bypass the EMI effect as an orthogonal measurement modality (see "Methods" section for the detailed experimental setup and Supplementary Fig. 10). As shown in Fig. 2e, we measured the pressure generated by the ImPULS at four different distances. The experimentally measured pressure matched well with the simulated pressure values. The slight discrepancy can be attributed to the minimum resolution of the 3-axis stage and the unknown true position of the sensing element in the fiber hydrophone tip. Next, we scanned the fiber-optic hydrophone in lateral x and y directions with a step size of 25 μm, keeping the axial z-distance constant at 15 μm. Figure 2f shows the pressure profile mapping, where a maximum pressure of 59.2 kPa is achieved at the center of the ImPULS. This characterization is important for the verification of precise and localized stimulation of neurons residing within the pressure field of the device.

## Durability, temperature stability, and biocompatibility

To test that an ImPULS remains functional over a long period of use within a harsh biological environment, we tested the durability of the ImPULS by performing an accelerated aging test in a phosphate-buffered saline (PBS) solution at an elevated temperature of 75 °C for 7 days (Supplementary Fig. 11). Figure 2g shows microscopic images of an ImPULS taken each 24 h apart, where there is minimal damage visually to the device after 7 days of continuous exposure to PBS at 75 °C. To confirm the ImPULS remains fully functional, we measured the displacement of the ImPULS before and after 7 days of aging test. As shown in Fig. 2g (right), the displacement of the device degrades only 2.4% in 7 days. In order to simulate the potential use of the ImPULS as a chronic device, we repeated the accelerated aging test with an additional application of stressor where a daily period of 10 min the device was turned ON. We recorded the voltage necessary each day to maintain the performance of the device as recorded on day 0. As seen in Supplementary Fig. 12, this adaptive voltage strategy enabled compensation for the loss of vibrational amplitude during the accelerated aging test, and the device sustained the displacement recorded at day 0 across 7 days.

To test the durability of the ImPULS further, we performed a fatigue test, during which a continuous sinusoidal signal of 500 kHz at 10 V (p-p) was applied continuously for 7 days and corresponding output displacement was recorded in a water medium. As shown in Supplementary Fig. 13, exposure to 302.4 billion cycles of a sine wave in 7 days results in a 40% lower amplitude of initial displacement, as degradation of the piezoelectric layer after extended electric cycling is a common phenomenon[42]. Compensation for performance loss due to piezoelectric degradation can be achieved by the application of adaptive voltage, the same strategy used in the accelerated aging test. We repeated the electromechanical fatigue test keeping the device ON for 5 days, and readjusted the driving voltage daily to achieve the target displacement equivalent to day 0 performance, as recorded by Laser Doppler Vibrometer (LDV). As seen in Supplementary Fig. 14, an adaptive voltage strategy was able to compensate for the expected displacement loss and could stabilize the performance of the ImPULS device.

Ultrasound generation from a transducer results in a temperature rise in the surrounding medium due to the intrinsic heating of the piezoelectric material and resistive losses[43]. We measured changes in temperature in a water medium (22.5 °C baseline temperature, Onda Aquas-10 tank) during ultrasound application using a dual sensing fiber-optic hydrophone capable of simultaneous measurements of acoustic pressure and temperature at the same location[44]. The fiber-optic thermometer positioned 15 μm away from the transducer recorded a temperature rise upon application of continuous ultrasound waves. As shown in Fig. 2h, 10 min application of continuous sinusoidal signal at an input voltage of 20 V (p-p) gave rise to only 0.6 °C which is much less than the threshold of temperature-evoked neuromodulation[10,45]. In practical neurostimulation applications, thermogenic effects are further reduced without affecting peak pressures generated due to the application of a pulsed ultrasound signal instead of a continuous signal. Further, we measured the temperature change upon application of pulsed signals with duty cycles of 50% and 5%. As shown in Supplementary Fig. 15a, b application of 50% and 5% duty cycle pulsed waves gave rise to only 0.15 °C and 0.03 °C, respectively. The dependence of temperature change on input voltage is shown in Supplementary Fig. 15c, indicating a maximum temperature change of 0.46 °C with an 18 V (p-p) input, which decreases to 0.08 °C with a 10 V (p-p) input. We also measured the temperature change in a 1.5% agarose gel (heat capacity of 3.9 J/kg/°C which is similar to brain tissue[46]) at the application of 20 V (p-p) continuous sinusoidal signal of 483 kHz, using a miniature beaded thermocouples (Evolution Sensors, Type K with bead diameter ~300 μm) for sensing. As shown in Supplementary Fig. 15d, e, the maximum temperature rises by 0.95 °C with the application of 20 V signal, which is less than the threshold of temperature-evoked neuromodulation[45]. In our ex vivo and in vivo neurostimulation experiments, a maximum of 10 V (p-p) is applied as will be described in later sections. Figure 2i shows the temperature profile mapping upon application of continuous sinusoidal signal where a maximum temperature change of 0.34 °C occurred, at 15 μm away in the z-direction from the probe center.

The ImPULS was surgically implanted in the deep brain for neurostimulation. To confirm that surgical implantation did not deteriorate its performance, we tested the ImPULS performance before and after insertion into the brain tissue-mimicking phantom. We prepared 0.6% agar gel to mimic the similar stiffness properties of brain tissue[47] and measured the displacement of ImPULS before and after insertion into the gel. As shown in Supplementary Fig. 16, the change in displacement is less than 1.5% confirming the device's stability after implantation into the brain tissue.

Next, the ImPULS device was assessed for the biocompatibility of its constituent materials with a cell viability test. Cortical tissues from embryonic mice were harvested and dissociated cells were seeded on glass-bottomed dished containing a fixed ImPULS device. The dissociated cells were allowed to differentiate into cultured primary neurons for a period of 10 days in the presence of the device. Cell densities across 6 plates (3 for control and 3 for ImPULS) were assessed and found to stabilize after a cell medium change on day 3 (Supplementary Fig. 17). Neurons on both Control and ImPULS plates differentiated neurites normally and developed neurites stably after the 10-day period.

## Hippocampal neuronal stimulation ex vivo

The ImPULS was evaluated for its potential to stimulate healthy neurons in a coronal hippocampal slice with two-photon imaging (Supplementary Fig. 18a). Hippocampal neurons expressed the genetically encoded calcium indicator GCaMP7F to report neural activity during ultrasonic stimulation[48]. Neurons in the dentate gyrus were targeted for stimulation (Supplementary Fig. 18b). After a 60 s baseline period, a sinusoidal pulse (500 kHz, 10 V(p-p) with 1.5 kHz pulse repetition frequency (PRF) and 50% duty factor) is used to stimulate neurons for 50 s. After stimulation ends, population activity is captured for another 60 s. Several neurons in the field of view were activated during ultrasound stimulation (Supplementary Fig. 18c and Supplementary

Video 1). Region of interest 1 (ROI 1) exhibits a 30% change in fluorescence approximately 15 s after stimulation begins and reaches maximum intensity 9 s after the initial rise. The other marked neurons belong to the same local cluster and showed smaller changes in activity during stimulation. The delay in neural activation could be due to high amounts of dissolved gasses in the artificial cerebrospinal fluid (aCSF), which may lower cavitation thresholds and cause ultrasound energy from small sources to be absorbed into the medium rather than the tissue[49]. In the absence of other stimuli, including the effects on active vasculature[50,51], these results confirm that ImPULS can activate local hippocampal neurons.

**Stimulation of CA1 in anesthetized mice induces cFos expression**
We next sought to test the potency of the ImPULS to activate cellular ensembles in vivo in mice. To do so, we surgically implanted the ImPULS in the hippocampus–a subcortical brain region that is essential for learning and memory across mammalian species[52] (Fig. 3a). Specifically, we targeted the ImPULS to the dorsal CA1 (dCA1) layer of the

hippocampal formation to test the efficacy of different stimulation protocols. We quantified the relative levels of cFos in dCA1, which is a widely used marker of recent neuronal activity, to measure the extent of neural activation resulting from different bouts of the ImPULS stimulation under anesthesia[53]. Compared to a no-stimulation (control) group, all stimulation parameters increased the levels of the activity-dependent gene cFos on average. Specifically, we observed an approximately 2-fold increase in cFos expression following stimulation with 500 kHz 10% duty factor (0.777 cFos+/mm² stim vs. 0.497 cFos +/mm² in control, one-way ANOVA), suggesting that this stimulation parameter was sufficient to activate large populations of cells in the dCA1 layer of the mouse hippocampus (Fig. 3b, c and Supplementary Figs. 19 and 20). We quantified significant increases in cFos expression along the entirety of dCA1. Furthermore, we visualized auxiliary cFos expression in CA3 and DG that could be due to backward propagating action potentials, local circuit increases in cFos resulting from ImPULS stimulation spread, and intra-hippocampal communication in general[54], which nonetheless underscore the potency of the ImPULS

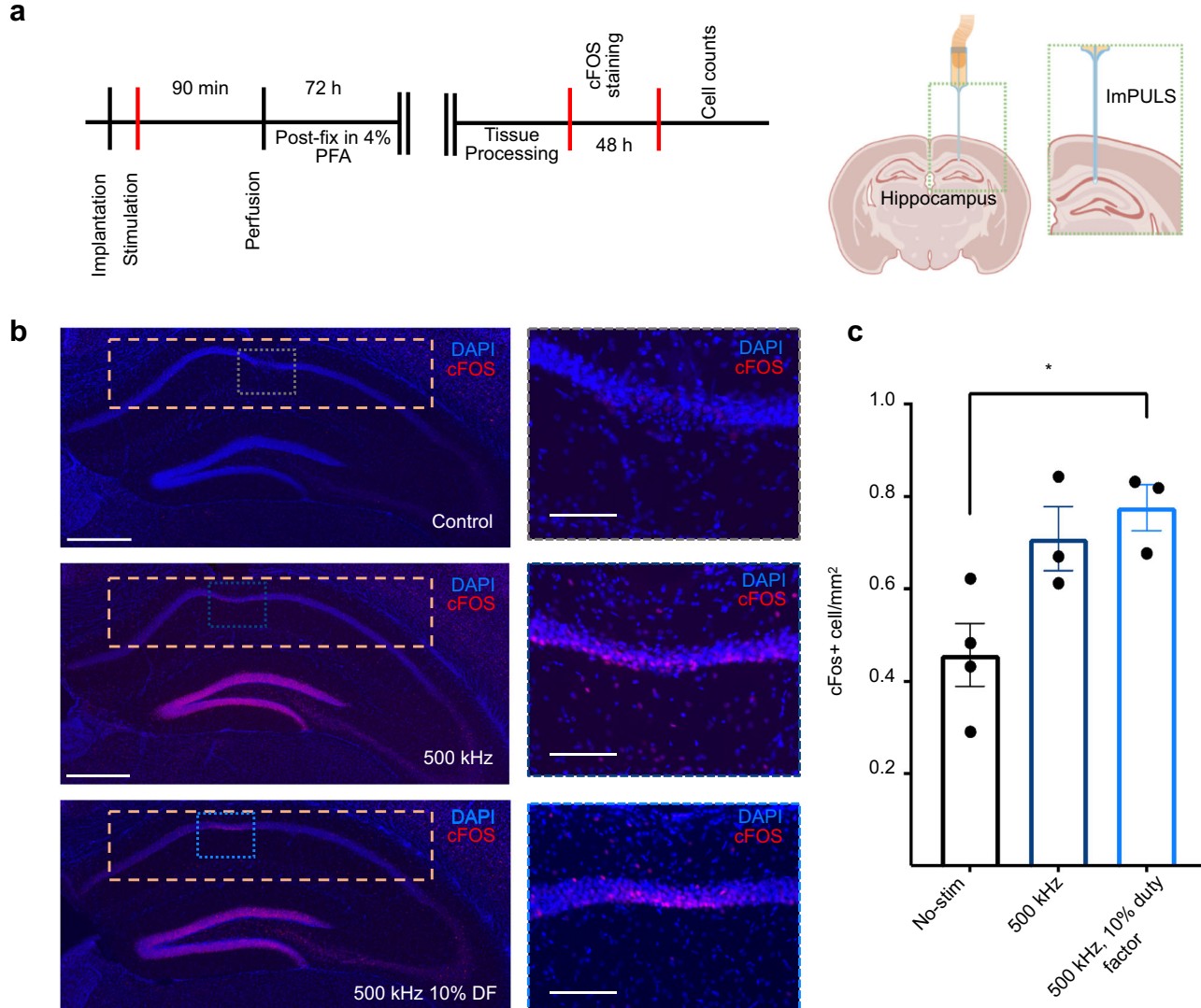

**Fig. 3 | Robust stimulation of the dCA1 in anesthetized mice. a** Experimental design and schematic diagram of surgical procedure. Schematic created with BioRender.com, released under a Creative Commons Attribution-NonCommercial-NoDerivs 4.0 International license. **b** Representative images of the hippocampus across experimental conditions: No-stim (top), and 500 kHz, 500 kHz, continuous wave for 60 s (middle), and 10% duty factor for 60 s (bottom). Yellow dashed boxed area approximates the dorsal CA1 (dCA1) area used for cell counts. Scale bar, 450 μm. Color-coded dashed box indicates the area shown in the magnified section. Scale bar, 100 μm. **c** cFos+ cells in dCA1 normalized by area across experimental conditions (One-way ANOVA; N = 4 mice control implant no-stim; N = 3 mice for 500 kHz condition; N = 3 mice for 500 kHz, 10% duty factor; No-stim vs. 500 kHz: p = 0.0506; No-stim vs. 500 kHz, 10% duty factor: p = 0.0184).

stimulation parameters. Furthermore, we tested ImPULS stimulation in the dCA1 layer on mice 14 days after ImPULS was implanted. The 500 kHz 10% duty factor stimulation group showed a significant increase in average cFos+/DAPI+ in the dCA1 layer using an unpaired *t*-test (*p* = 0.0258). Our results indicate that ImPULS stimulation was sufficient to elicit neuronal activation following chronic implantation. This suggests the ImPULS device is functionally viable on more chronic timescales. Immunohistological staining of GFAP demonstrated minimal microglial activation in response to chronic implantation in the area of tissue most proximal to the ImPULS transducer (Supplementary Fig. 21). Together, these results demonstrate the applicability of in vivo neuronal stimulation using an ImPULS.

## Ultrasound stimulation of the SNc modulates nigrostriatal dopamine release in an anesthetized mouse

Next, we tested the utility of an ImPULS stimulation for functional modulation of neurotransmission in vivo using an anesthetized preparation. Dopaminergic (DA) neurons of the SNc innervate the dorsal striatum (DS) to form the canonical nigrostriatal dopamine pathway, a circuit crucial for movement and reinforcement in the mammalian brain. Furthermore, excitatory stimulation of dopaminergic transmission has therapeutic implications in the treatment of Parkinson's Disease (PD) and memory disorders[55,56]. Therefore, we sought to modulate nigrostriatal DA release through the ImPULS stimulation of the SNc using an anesthetized preparation (Fig. 4a). We targeted the

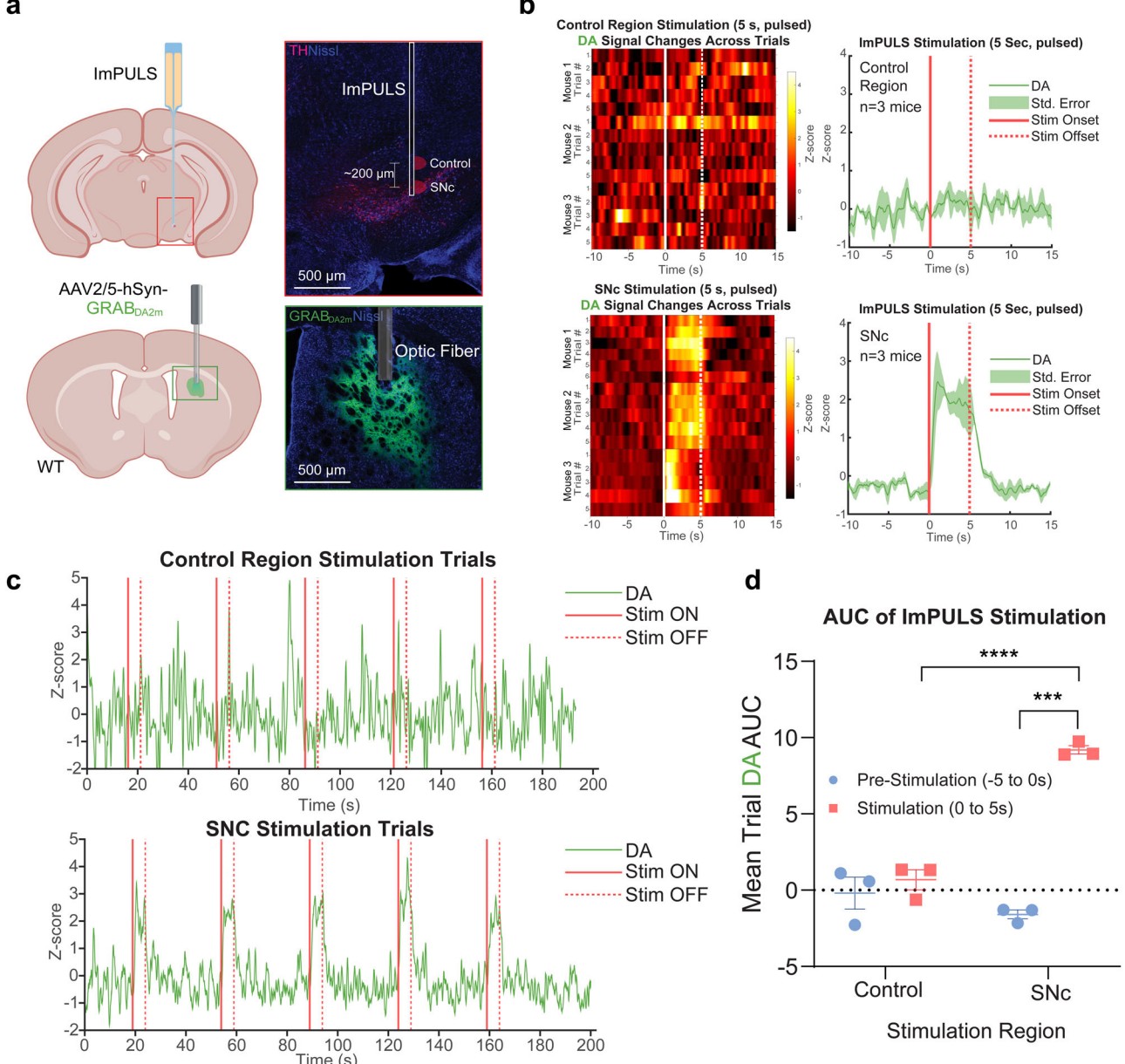

**Fig. 4 | Stimulation of nigrostriatal dopamine release in anesthetized mice. a** Schematic diagram of the experimental approach to stimulate substantia nigra pars compacta (SNc) dopaminergic (DA) neurons, including post-hoc histological validation of on-target implantation and DA2m sensor expression. Scale bar, 500 μm. Schematic created with BioRender.com, released under a Creative Commons Attribution-NonCommercial-NoDerivs 4.0 International license. **b** Averaged DA2m fluorescence responses for control (top right) and SNc (bottom right) stimulation trials. Average heatmaps of fluorescence across trials for control (top left,

3 mice) and SNc (bottom left, 3 mice) stimulation trials. Solid line data represents mean data and bands represent ± SD. **c**, Full recording trace of Z-score DA2m fluorescence across stimulation trials, with onset and offset of stimulation (5 s,1500 Hz, 50% duty factor) indicated by solid and dashed red lines, respectively. **d**, Area under the curve (AUC) analysis for 5 s pre-stimulation versus 5 s during stimulation for average control and SNc stimulation trials (3 mice/group). Repeated-measures 2-way ANOVA with Sidak's multiple comparisons test (***p* = 0.0009, *****p* < 0.0001).

ImPULS implantation to the anterior SNc with a lateral stimulation direction and performed fiber photometry recordings of extracellular DA release in the DS using a genetically encoded DA sensor, GRAB-DA2m[57]. Further details describing the surgical approach, recording parameters, and data analysis can be found in the Supplementary Information. Pulsed (PRF 1500 Hz, 50% duty factor) stimulation of the SNc for 5 s (514 kHz, 10 V(p-p)) elicited robust, time-locked increases in striatal DA release (Fig. 4b). Notably, control stimulation trials, in which we stimulated tissue approximately 200 μm dorsal to the SNc, failed to alter DA2m fluorescence (Fig. 4c, top and Supplementary Fig. 22). Therefore, at least in the areas of tissue inferior to the device, stimulation does not reach beyond 100 μm. In contrast, we observed a mean increase in DA2m fluorescence between 2 and 3 Z-scores throughout the duration of ImPULS-mediated SNc stimulation (Fig. 4c, bottom). As seen in the raw traces, reduced magnitude and variability of DA signaling is pronounced between stimulation and control trials. We calculated the mean area under the curve (AUC) across stimulation trials from the 5 s baseline period before stimulation and 5 s period during stimulation of both SNc and control tissue (Fig. 4d). Mean AUC of DA fluorescence during SNc ImPULS stimulation was significantly different from baseline ($F_{1,4} = 65.20$, $p = 0.0013$) using a two-way repeated measures ANOVA. Furthermore, SNc stimulation was significantly different from control stimulation ($F_{1,4} = 39.46$, $p = 0.0033$), with an interaction present between these two conditions ($F_{1,4} = 47.00$, $p = 0.0024$). Šídák's test for multiple comparisons showed that DA AUC during SNc stimulation dramatically increased compared to pre-stimulation baseline ($p = 0.0009$), with no difference between baseline and stimulation AUC for control tissue ($p = 0.6834$). Furthermore, DA AUC during SNc stimulation was significantly greater than control tissue stimulation ($p < 0.0001$) with no significant difference in baseline AUC between SNc and control tissue stimulation. Taken together, these data suggest that the ImPULS evokes robust nigrostriatal DA release in a spatially localized manner, given that the stimulation of tissue only 200 μm above the SNc failed to alter DA release. In four mice, we observed a mean increase in DA2m fluorescence between 3 and 4 Z-scores throughout ImPULS-mediated SNc stimulation for 1.5 s with these same parameters (Supplementary Fig. 23a), indicating that stimulation of shorter durations is sufficient to induce DA release. In a fifth mouse, however, we observed a mean decrease in DA2m fluorescence between 2 and 3 Z-scores throughout ImPULS-mediated stimulation for 5 s with these parameters (Supplementary Fig. 23b). Sections of the SNc were stained for tyrosine hydroxylase (TH), a widely used marker of dopamine-producing cells, to validate that the ImPULS probe tract contacted dopaminergic neurons in the SNc (Fig. 4a). There are multiple plausible explanations for the inhibition of nigrostriatal DA release. First, ImPULS may elicit both excitatory and inhibitory effects through an unknown underlying mechanism. Second, this may result from the surgical targeting of the SNc with a small and flexible probe. The substantia nigra pars reticulata (SNr), which is predominantly composed of GABAergic neurons, is situated directly ventral to the SNc and provides monosynaptic inhibitory input to DA neurons[58]. Given that the SNc DA neurons are angled on the medial-lateral axis and the direction of ImPULS stimulation was lateral, it is feasible that stimulation reached a significant number of SNr inhibitory neurons in this particular case, effectively silencing DA neurons. Additionally, the tip of the probe tract for this case reached the ventralmost portion of the SNc. Finally, a third explanation for these different effects of ImPULS stimulation is the possibility that deep brain ultrasound stimulation may differentially modulate neural activity by indirect modulation, affecting the activity of astrocytes[13]. Nevertheless, these data demonstrate that spatially localized deep brain ultrasound stimulation is capable of modulating neurotransmission in vivo, even through long-range projections.

## Discussion

This work presents a micron-sized, implantable ultrasound stimulator capable of modulating neuronal activity in deep subcortical regions and nigrostriatal dopamine production across long-range projections. We demonstrated the scalable microfabrication of an ImPULS, including the use of biocompatible materials such as the active piezoelectric element (KNN), interconnects, and encapsulation, as well as control over resonance frequency within the range of 0.2–1 MHz through manipulation of cavity size. We demonstrated that ImPULS has consistent resonant behavior (Supplementary Fig. 5) and minimal heating <1 °C (Supplementary Fig. 15) while placed in different acoustic and thermal tissue-mimicking mediums such as 1.5%, 0.6% agar gel, and air. When operating at its resonant frequency, the ImPULS drew 0.2 mA current with 10 V (p-p) of applied voltage. At a power consumption of 0.2 mW, the ImPULS generated ultrasound pressures of 100 kPa at resonance frequency in pulsed and continuous waves within its stimulation region and evoked modulation of cell activity without causing thermogenic effects on nearby cells. The modulation of brain circuitry with a pressure of 100 kPa around 500 kHz is consistent with prior literature of in vivo experiments with mice using similar ultrasound driving parameters[21]. The ImPULS elicited neuronal excitation in the hippocampus ex vivo, induced activity-dependent gene expression in hippocampal cells of an anesthetized mouse, and modulated dopaminergic neurons in the SNc to elicit precise timing of striatal dopamine release. This presents the ImPULS as a potent neuromodulatory tool.

Cells within the hippocampus have been shown to be sensitive to ultrasound stimulation from transcranial and slice preparations[48]. Within our immunochemistry studies of the ImPULS stimulation in the hippocampus, the ImPULS activated large populations of neurons in upstream canonical pathways relative to where stimulation occurred. For instance, cells in the dentate gyrus are active accompanying stimulation in the CA1 region. Contained within the intrinsic circuitry of the hippocampal formation are several parallel processing and feedback networks mediating excitation, inhibition, and disinhibition[59]. Previous research has shown that inducing hypersynchrony in CA1 using optogenetic stimulation was sufficient to activate the entire hippocampal formation[60], which was consistent with our work showing increased cFos expression in CA1, the DG, and CA3.

We achieved robust stimulation using an ultrasound-driving protocol that is known to excite neurons[22,61], but we cannot discard the possibility that there is an indirect pathway for neuronal excitation via astrocytic modulation. Oh et al. [13] demonstrated in co-cultures of neurons and astrocytes that TRPA-channels from astrocytes can be ultrasonically activated and are sufficient to indirectly excite neurons via glutamate release[13]. Demonstrations that illustrate the spatial resolution of an ImPULS would be better replicated in regions of the brain where fewer interconnections between neural circuits exist and further studies investigating different stimulation parameters could potentially be used to achieve a degree of cell selectivity as demonstrated by other groups[62].

Implantable ultrasound neurostimulation offers several advantages compared to other neurostimulation modalities. Unlike electrode-based deep brain stimulation, the ImPULS has no exposed electrochemical area, thus avoiding biofouling and corrosion, and compared to the DBS cross-sectional footprint (in the mm range) and rigid form factor, the ImPULS is much smaller yet still capable of precise and potent simulation. Furthermore, as glial scar tissue forms in response to the implant, the acoustic properties of the encapsulating tissue change negligibly. Therefore, it retains desirable properties as the propagating medium for the effective delivery of ultrasound energy. The ImPULS is implanted adjacent to target neurons, the ultrasound generated bypasses the skull and is focused to a volume <100 μm³, thereby avoiding off-target activation from scattering and reflections. Finally, the use of ultrasound for neurostimulation offers

the prospect of non-genetic and cell-specific modulation through its large stimulation parameter space, which has been demonstrated in recent studies[63] by selecting different pulse repetition frequencies (PRF). Auditory confounds in transcranial ultrasound-based stimulation are widely reported and often addressed through strategies such as genetic deafening of mice[23]. However, they can also be circumvented altogether through the use of precise microfabricated implantable devices, which have precise and compact focal volumes and bypass transmission through or near the skull.

Future studies can gain finer control of stimulation and evaluate potentially distinct effects of ImPULS-mediated stimulation (i.e., excitation vs. inhibition) on different cell types, neural circuits, and regions as indicated by existing studies[64–66]. The current ImPULS device is not able to produce pressures higher than 100 kPa due to the polarization saturation limits of the piezoceramic thin film, but the ImPULS is able to produce neuromodulatory effects in distinct regions of the brain. Device iterations that can reach higher pressures with multiple elements or improved piezoelectric performance can be leveraged for more efficient and widespread stimulation. The durability of the ImPULS is expected to survive the duration of a month-long chronic implantation based on the results of our accelerated aging tests, and changes in surrounding tissue stiffness during chronic implantation have been simulated to affect the resonant frequency of an ImPULS by less than 8% percent, which can be sensed and matched by impedance sensing electronics. Further developments and investigations would be towards device stability, connected electronics to facilitate chronic behavioral studies, and integration with sensing elements for achieving responsive stimulation. It is evident that ultrasound neuromodulation has the potential to address unmet needs in the treatment of diseases in the deep brain, and for this potential to be realized, a thorough parameter space investigation covering different ultrasound driving parameters and piezoelectric architectures would be valuable future work. By using the targeted stimulation capability of the ImPULS together with different acoustic parameters, we believe this implanted ultrasound stimulation device can be developed into a versatile tool for both basic systems neuroscience research and potential therapeutic applications.

## Methods

### Piezoelectric properties of KNN
The KNN characterization curve (Supplementary Fig. 4) was measured by Sumitomo Chemicals Co. (Japan) to assess the polarization hysteresis loop and electromechanical efficiency of the piezoelectric material[67]. These properties were measured by a double beam laser interferometer under an applied ±30 V, 1 kHz sine wave voltage, and a 0.5 mm diameter top electrode. The larger dielectric constant of KNN enables higher piezoelectric displacement which is ideal for actuator-type applications such as ultrasound stimulation transducers.

### The ImPULS Fabrication
A starting thin-film stack of 500 nm $SiO_2$/610 µm Si/500 nm $SiO_2$/ 30 nm ZnO/200 nm Pt/1 µm KNN/10 nm Cr/150 nm Au on a donor wafer (SCIOCS Co. Ltd., Sumitomo Chemical Group, JAPAN) was processed in preparation for transfer printing. Contact photolithography (Karl Suss MJB4) with spin-coated positive i-line photoresist (PR) (Microchemicals AZ 4620) of 15 µm thickness was used to define wet-etching patterns for top and bottom electrodes and the piezoelectric transducer. Top electrode Au and Cr layers were wet etched with gold etchant (Transene) and CR-7 chrome etchant (Transene), respectively. KNN was subsequently defined using the same photoresist mask and wet etching with a 49% hydrofluoric acid (HF, JT Baker). We introduce an alternative acidic wet-etching chemistry for KNN patterning that can achieve faster etch rates of up to an etch rate of 100 nm/min, which mediates the undercut effect and allows for denser and smaller transducer pitches. PR was then stripped in a 120 °C heated NMP-based

stripper (Microchem AZ400T). The Pt bottom electrode pattern was defined with a PR etch mask (AZ 4620) and both ZnO/Pt were dry-etched using a reactive-ion etching (RIE) flowing Ar/$O_2$ (95%/5%) and power of 500 W (Plasmatherm). A final single PR layer was designed to both preserve element-to-element alignment and serve as a mechanical support during undercutting, release, and transfer printing. The thin-film stack was finally undercut with diluted hydrogen fluoride (HF) solution with a weight ratio of 49%/60%: HF/Deionized (DI) water, rinsed, and delaminated from remaining thermal oxide using thermal release tape (Revalpha 90 °C). This anchor-layer step is crucial to maintaining the relative spacing of elements during the HF undercut. We designed a special anchor modifying our previously reported recipe[68] to extend the processing window and mechanically support the pattern with PR ensuring high yield in transfer to thermal tape (see Supplementary Fig. 2 for details). The patterns were transferred to the tape and prepared to be printed on an SU-8 (Kayaku Advanced Materials) flexible substrate.

A separate substrate Si wafer was coated with an Omnicoat (Kayaku Advanced Materials) release layer, followed by a 0.5 µm thick layer of SU-8 2000.5. The wafer was then soft-baked and flood-exposed for 2 s with 365 nm ultraviolet (UV) light. The collected pattern on the thermal release tape was then pressed onto the substrate and left on a hot plate to slowly heat up to release temperature during the post-exposure bake of the SU-8. Finally, at the release temperature, the tape was gently removed with tweezers and the PR anchor was removed with sprayed acetone and 2-propanol. Following a short RIE $O_2$ plasma treatment, an insulation and opening layer of SU-8 was spun onto the wafer and patterned to leave opening contacts to the top and bottom electrodes for metallization and metal interconnects to bond pads. The entire structure was then hard-baked at 120 °C to ensure proper adhesion between layers and reduce the angle of the sidewalls around the metal contact openings to ensure proper connectivity after metallization. 10 nm of Cr and 200 nm of Au were sputter (AJA ATC Orion 5) deposited over the entire substrate and patterned with a PR mask for wet etching to define the metal interconnects and larger bond pads that will allow the device to be connected to external instrumentation. After the device is rinsed and the resistivity of the resulting surface is checked to ensure complete etching of conductive material, the device is stripped of PR, cleaned, dried with $N_2$ spray gun, and treated with $O_2$ plasma preceding deposition of the next insulating layer. Next, a 0.5–0.8 µm thick layer of SU-8 is spun, exposed, and cured to complete the insulation of the electrically active elements.

The cavity layer is a thicker mechanical layer (~15 µm) that can be designed to control the resonant frequency of the device. The dimensions of the cavity determine the resonance frequency (Supplementary Fig. 6a.). The ImPULS requires sufficient stiffness for ease of surgical implantation without the need for an insertion shuttle. We fabricated an SU-8 backing layer of ~15 µm to give the device sufficient stiffness while maintaining the overall flexibility. First, a thin layer of polyvinyl alcohol (PVA) (4 wt%) was spin-coated to a polyethylene terephthalate (PET) film at 3000 rpm for 45 s followed by an annealing process of 10 min at 110 °C. This PVA layer works as a sacrificial layer to aid the delamination of the PET backing during development. Following a short RIE $O_2$ plasma treatment, SU-8 2050 was spun at 6000 rpm for 45 s. The sample was then soft-baked for 5 min at 65 °C followed by 5 min at 95 °C in order to achieve an appropriate surface for bonding. The previously prepared wafer with the cavity layer and the rest of the device structures were then bonded to the SU-8 on PVA and pressed gently. The final pattern which defines the device perimeter was then exposed through the PET film to seal the cavity layer and stiffen the device. After a post-exposure bake, the bonded substrates were soaked briefly in 2-propanol to dissolve the PVA sacrificial layer. Finally, the unexposed SU-8 was developed away.

In order to release the devices from the wafer substrate, the bottom-most layer of SU-8 was blanket dry-etched in $CF_4/O_2$ for a

minute to expose the initial Omnicoat layer and release the devices. Once the SU-8 was etched, the devices were immersed in MF-26A developer (Kayaku Advanced Materials) for release. The devices were allowed to soak for 45 min for the Omnicoat to dissolve completely and were then rinsed in 2-propanol and DI water until clean. The devices were collected and arranged under a final shadow mask and dry-etched to expose the metal bond pads, which were used to connect to the cable.

## Scanning electron microscopy (SEM)
The samples were prepared by cutting the transducer of ImPULS using a sharp razor blade under a microscope (10× magnification). 10 nm Au was then sputter deposited on the sample to mitigate charge accumulation (Balzers Union SCD 040). The SEM observation was performed using Zeiss Gemini 450 SEM with an excitation voltage of 3 kV.

## Assembling of ImPULS
The PCB was designed using Altium Designer. ACF cable (Elform Inc.) and used to connect ImPULS to a PCB. A heat-press using a solder tip at 180 °C ensured conformal bonding between cable and contact pads of both the device and PCB, which was further confirmed using a multimeter and electrical impedance spectroscopy. The ACF cable was then finally encapsulated using Kapton tape.

## Electrical impedance spectroscopy & transducer power consumption
An impedance analyzer (E4990A, Keysight Technologies) was used to characterize the electrical impedance spectrum to determine the resonant frequency in both air and water mediums. In order to measure the impedance characteristics of the microfabricated device alone, the system was calibrated to probe tip terminations that were used to contact a Gallium Indium eutectic (Sigma Aldrich) liquid-metal bridge extending from the contact pads of the device.

The power consumption of ImPULS can be determined through impedance measurements of the assembled device at the resonance frequency and the voltage applied through the following equation: $P_{avg} = 1/2 \ V_{max}I_{max}\cos(\theta_i - \theta_v)$. With 10 V(p-p), the ImPULS device without cabling consumes 1.3 mW of power.

## Electromechanical characterization
The displacement amplitude of the device was measured using a Laser Doppler Vibrometer-based non-contact vibration measurements (LDV, MSA-500, Polytec). The device was mounted on a stainless-steel chuck, and an electrical AC driving signal was applied to the device using the system's internal function generator. The resonant frequency spectrum was determined by applying a periodic chirp signal. Later, a sinusoidal signal is applied at the resonant frequency to measure the maximum vibration amplitude. This experiment was conducted in both air, water, and agar gel mediums.

## FEA simulation
COMSOL Multiphysics (version 6.0) was employed for the simulation of acoustic pressure generated in a water medium by ImPULS. The materials properties of KNN are used as follows: density of 4000 Kg/m³, Young's modulus of 65 GPa, relative permittivity of 1500, and piezo constant of $e_{31}$ 12 C/m². The model uses finer mesh and solves the pressure acoustics, electrostatics, and solid mechanics physics for the solution. The geometry of the model matches the dimensions of experimental geometry.

## Acoustic and thermal characterization
The acoustic characterization was performed by measuring the pressure emitted from ImPULS using a fiber-optic hydrophone system (Precision Acoustics), based on the detection of acoustically and thermally-induced thickness changes in a polymer film Fabry–Pérot interferometer deposited at the tip of a single mode optical fiber. This system enables simultaneous temperature and pressure measurements across a circular sensing area of 100 μm in diameter. The ImPULS was driven by a benchtop function generator (BK Precision) with a 20 V pulse sequence.

In order to perform a 2-D mapping of pressure and temperature across the surface of the device, the fiber-optic hydrophone was mounted to a 3-axis micromanipulator (Newport), and the tip was directed into a petri dish containing deionized and degassed water at 22 °C. The ImPULS device was fixed to the bottom of the dish. Under microscope magnification, the tip of the hydrophone was steered into view and aligned on top of the device as seen in Supplementary Fig. 10. The manipulator was advanced using manual manipulators, and distances were measured using the microscope images taken during the experiment. After the hydrophone was driven into position, ImPULS was turned on, and the generated voltage waveform from the hydrophone was collected by an oscilloscope (Agilent, 100 MHz 4 GSa/s) at each location.

Axial pressure measurements were conducted in an identical setup with ImPULS rotated 90° about its longer axis. Therefore, when the hydrophone tip was manipulated to the vicinity of ImPULS, it was oriented to take pressure measurements in a direction normal to the active surface of the ImPULS device. We then manually advanced the device away from the hydrophone in the plane, measured distance with magnified images taken by a microscope camera mounted above the device and hydrophone, and took a series of pressure measurements (Supplementary Fig. 10). We approximated an error margin in the measured distance due to the partial occlusion of the sensing area on the hydrophone.

Thermal characterization in 1.5% agar gel phantom was performed by measuring the local temperature near the transducer surface with simultaneous determination of the device's vibration amplitude. An ImPULS device was mounted on a dish underneath a 10× objective. A miniature beaded thermocouple (Evolution Sensors, Type K with bead diameter ~300 μm) was attached to a micromanipulator, and the bead was moved into proximity of the ImPULS surface. Agar gel with 1.5% was prepared and drawn into a pipette while in the liquid state and dispensed on top of the thermocouple and ImPULS device and allowed to cool and gel. Laser doppler vibrometer measurements were taken to determine the resonant frequency and operating characteristics of the device in gel (Polytec MSA-500) and temperature measurements were acquired at 25 Hz (DataQ Instruments DI-245).

## Accelerated aging test
The accelerated aging tests (with constant and adaptive voltage) were conducted by inserting test devices and a thermistor inside a beaker containing 10× phosphate-buffered saline (PBS) solution (Sigma Aldrich). The beaker was placed on top of a hot plate and the temperature was maintained at 75 °C. Because there is a temperature difference between the stage of the hot plate and the PBS inside a beaker, we adjusted the hot plate temperature to 95 °C and used a thermistor to check the inside temperature to remain constant at 75 °C. The beaker was capped with a 3-D-printed cap to ensure the PBS did not evaporate during the test time. The PBS solution was replaced with fresh PBS after three days. Microscopic images of the devices were taken daily. For the aging test with constant voltage, the displacement of the device was measured before starting the experiment and at the end of the 7th day using a laser Doppler vibrometer as explained above. For the aging test with adaptive voltage, the maximum device membrane displacement under actuation was measured using a laser Doppler vibrometer on day 0, and the target displacement of 120 nm under 2 V was achieved across 3 devices. On subsequent days, the device was turned on for 10 min at the resonant frequency, with a 50% duty cycle and 1.5 KHz PRF while submerged in PBS to add corrosive stress. The applied voltage was incrementally increased until it reached

the max membrane displacement reached 120 nm and the voltage was recorded. Three devices were tested for the accelerated aging test with constant voltage and three for the aging test with adaptive voltage.

## Electromechanical fatigue test

The device fatigue test with constant voltage was conducted by submerging probes in water and continuously applying a sinusoidal signal (10 V p-p) at their respective resonant frequencies. The probes were kept submerged under water and displacement was measured every day at a fixed time for 7 days. The devices used for this test had cavities of 105 μm and a resonant frequency of 500 kHz. Therefore, each device was exposed to 302.4 billion cycles of the sinusoidal signal. The electromechanical fatigue test was repeated using an adaptive voltage strategy, taking the measured displacement of 120 nm at the Laser Doppler Vibrometer (LDV, Polytec), equivalent to a driving voltage of 2 V on day 0, as the target displacement to be reached by readjusting the voltage throughout 5 days in case of performance loss.

## Insertion test

The insertion test was performed by inserting devices into agarose gel and checking the change in displacement of each device. The brain tissue has a similar stiffness of 0.6% agar gel[47]. We prepared an agar gel with 0.6% concentration by mass. The device was mounted on a 3-D stage and lowered manually to insert into the agar gel. After insertion, we kept the devices inside the gel for 5 min before removing the device from the gel. The displacement was measured before insertion and after removal using LDV.

## Animals

All animal handling and experiment procedures were approved by The MIT Committee for Animal Care, the Institutional Animal Care and User Committee of the California Institute of Technology, and the Boston University Institutional Animal Care and Use Committee. Adult female C57BL/6 mice were used in this study for SNc experiments, adult male C57BL/6 were used for hippocampus and slice experiments (Jackson Labs, stock number 000664), and embryonic day 17 C57BL/6J mice (The Jackson Labs) were used for cell culture experiments.

## Primary neuron preparation and cell viability testing of cultured primary neurons on ImPULS devices

All animal procedures were approved by the Institutional Animal Care and User Committee of the California Institute of Technology. Cortical tissues were dissected from embryonic day 17 C57BL/6J mice (The Jackson Laboratory). Cell culture dishes were prepared from 3.5 cm diameter glass-bottom dishes (35 pi, Matsunami, Osaka, Japan) and coated with poly-D-lysine (0.1 mg/ml, Gibco) overnight, and washed with deionized water and dried with vacuum aspiration. The extracted tissues were rinsed with Hank's Balanced Salt Solution (VWR) and dissociated by pipetting, followed by centrifugation at $200 \times g$ (1500 rpm) for 2 min. Pellets were collected and re-suspended in the culture medium. Cells were seeded at an approximate density of 300–1000 cells/mm². Neurons were cultured on a glass dish and maintained in 2 mL of Neurobasal medium (Thermo Fisher Scientific) supplemented with B27 (2% v/v, Thermo Fisher Scientific), GlutaMax (2 mM, Gibco), glutamate (12.5 μM, Sigma) and penicillin/streptomycin (1% v/v, Corning) in a humidified incubator with 5% CO2 and 37 °C. Half the medium was replaced with fresh medium without glutamate every 3 days.

3 control plates without an ImPULS device present and 3 plates with an ImPULS device present were imaged over a period of 10 days. The ImPULS device was mounted to the bottom of the glass dish with a drop of cyanoacrylate adhesive several millimeters distal to the ImPULS tip. Phase contrast images of a representative area in the plate were taken on a PrimoVert inverted microscope at a magnification of 10× in the air (Zeiss A-Plan; 421041-991; 10×/0.25 Ph1; inf/-). Cell bodies

were counted manually across a 135 μm × 135 μm representative tile using ImageJ (Supplementary Data).

For immunostaining, primary neurons were fixed using ice-cold paraformaldehyde (4% in PBS, VWR) and 5% sucrose in PBS for 20 min at 4 °C and washed 3 times with PBS. Cell membrane was permeabilized with 0.1% Triton-X100 in PBS for 5 min, followed by PBS wash. Non-specific binding was blocked by 1% bovine serum albumin (Sigma) and 1% normal goat serum (Abcam) in PBS for 30 min at room temperature and cells were after washed with PBS. Primary antibody (anti MAPS2 (1:500, Abcam)), was diluted in 1% bovine serum albumin and 1% normal goat serum and incubated with cells for 1 h at room temperature. After washing with PBS 3 times, secondary antibody (Alexa Fluor 488 (1:500, Abcam)) diluted in 1% bovine serum albumin and 1% normal goat serum were loaded to neurons for 1 h at room temperature and kept in the dark. After washing with PBS 3 times, cells were imaged using a confocal microscope (LSM 980 with Air scan, Zeiss).

## Calcium imaging in brain slice

Acute brain slices were prepared to measure calcium dynamics during ultrasound stimulation ex vivo. All experiments were conducted in compliance with Boston University Institutional Animal Care and Use Committee (IACUC) approved protocol 201800599. Adult C57BL/6J mice (JAX, strain #000664) of either sex were intracortically injected with an adeno-associated virus for non-specific expression of GCaMP7f (AAV9-syn-jGCaMP7f-WPRE) in the hippocampus at coordinates −2.0 AP, −1.4 ML, −2.0 DV. Acute coronal slices of the hippocampus were prepared at least 3 weeks following injection. After anesthetization with isoflurane and decapitation, brains were extracted and immersed in 0 °C sucrose-substituted artificial cerebrospinal fluid (in mM): sucrose (185), KCl (2.5), NaH₂PO₄ (1.25), MgCl₂ (10), NaHCO₃ (25), Glucose (12.5), CaCl₂ (0.5). Slices were cut to a thickness of 250 μm (Leica VT 1200, Leica Microsystems). Slices were then incubated at 35 °C for 30 min in artificial cerebrospinal fluid (ACSF) consisting of the following (in mM): NaCl (125), NaHCO₃ (25), D-glucose (25), KCl (2), CaCl₂ (2), NaH₂PO₄ (1.25) and MgCl₂ (1). Afterwards, the slices were cooled to room temperature (20 °C). After the incubation period, slices were moved to the stage of a two-photon imaging system (Thorlabs) with a mode-locked Titanium:Sapphire laser (Chameleon Ultra II; Coherent) set to a wavelength of 910 nm to excite GCaMP7f using a 20×, NA 1.0 (Olympus) objective lens. Laser scanning was performed using resonant scanners and fluorescence was detected using a photo-multiplier tube (Hamamatsu) equipped with a green filter to record emission from GCaMP7f. The stage of the microscope contained recirculating ACSF perfused with 95% O₂/5% CO₂. The temperature of the bath was heated between 35–37 °C. A video with 1365 frames and a duration of 170 s was captured with a frame rate of 8.02 frames per second.

Fluorescence emission intensities were extracted from the captured video and processed by the CaImAn Python package. Frames in the video are first processed for motion correction, which was negligible due to the fixed nature and minimal drift of the slice. Next, a constrained non-negative matrix factorization (CNMF) algorithm is used for source extraction and deconvolution to extract the spatial and temporal components within the frames. Finally, components are automatically evaluated based on the signal-to-noise ratio (SNR), spatial correlation, and neuron shape likelihood (based on a convolutional neural network) of the segmented components. Significant fluorescent traces are manually chosen and the background-normalized change in fluorescence is calculated for each neuron to determine the $\Delta F/F_0$ by percent.

## In vivo stimulation in anesthetized mice and immunohistochemistry

We devised an experimental paradigm to evaluate tissue activation resulting from ultrasound stimulation (US). All experimental protocols were approved by the Institutional Animal Care and Use Committee at Boston University. Adult male C57BL/6 underwent stereotactic

implantation of the ImPULS device into the dCA1 of the hippocampus at the following coordinates: coordinates AP −2.00 mm, ML + 1.40 mm, DV −1.50 mm. When these coordinates were reached, 5 min elapsed before stimulation to allow the probe to settle in the surrounding tissue. The ImPULS was positioned to face posteriorly in the anterior-posterior axis so the probe tract could be visualized. The device was secured using dental cement and subjects were allowed to recover before timed perfusions.

All mice were perfused 90 min following ultrasound stimulation to capture peak cFos expression. Mice were first overdosed with iso-flurane before undergoing transcardial perfusions with 4 °C phosphate-buffered saline (PBS) followed by 4% paraformaldehyde (PFA) in PBS. Intact brains were postfixed with the probe still inside for 72 h to facilitate visualization of the probe tract. Tissue sections (50 μm thickness) were then collected using a Leica VT1000 S vibratome equipped with a platinum-coated double-edged blade (Electron Microscopy Sciences, Cat. #72003-01) and set to a maximal speed of 0.9 mm/s. Sections were chosen based on their proximity to the implantation site - where the probe tract could be directly visualized and those directly anterior and posterior to it. Sections were stained for cFos[69] and then mounted onto micro slices (VWR International, LLC). Vectashield HardSet Mounting Medium with DAPI was applied and slides were coverslipped. Slices were stained with primary antibodies 1:1000 rabbit anti-cFos (Synaptic Systems), 1:1000 chicken anti-GFP (Invitrogen), and secondary antibodies 1:200 Alexa 555 anti-rabbit (Invitrogen); 1:200 Alexa 488 anti-chicken (Invitrogen). All slides were given 12 h to dry at room temperature before imaging. Slices were imaged utilizing an Akoya Biosciences Vectra Polaris Imaging System at 20× magnification. Images were then aligned to the Allen Brain Reference Atlas to determine the bounds of the dCA1 area and cropped accordingly. Cell counts were calculated for the entire dCA1 layer across the medial-lateral axis of the implanted hippocampus. cFos positive cells were quantified using QuPath, a machine-learning-based bioimage analysis pipeline used to classify regions of interest[70]. Histological images were chosen at random to serve as training data in the QuPath program. cFos positive cells were identified and counted by the algorithm with the experimenter blind to treatment and context groups. Counts for cFos-positive cells were normalized to the total area of the ROI (counts/mm²).

## Chronic implantation of ImPULS, terminal stimulation, and immunohistochemistry of GFAP and cFos

We devised an experimental paradigm to evaluate tissue activation resulting from ultrasound stimulation (US) after a period of 14 days. All experimental protocols were approved by the Institutional Animal Care and Use Committee at Boston University. Mice underwent stereotactic implantation of the ImPULS device into the dCA1 of the hippocampus at the following coordinates: coordinates AP −2.00 mm, ML + 1.40 mm, DV −1.50 mm. The ImPULS was positioned to face posteriorly in the anterior-posterior axis so the probe tract could be visualized. The device and headcap base were secured using dental cement and the device and ribbon connector were detached from the stereotaxic frame by severing temporary connections. A removable headcap[71] was placed to protect the device connector and subjects were allowed to recover before being placed in a cage for long-term keeping.

After a period of 14 days, adult male C57BL/6 mice were anesthetized, and the ribbon connectors were uncapped and reconnected to a PCB connector for alternating voltage input. The impedance of the device was checked to ensure stable reconnection to the PCB and viability of the piezoelectric element. Mice were then stimulated for 60 s with 500 kHz, 50% duty cycle, and 1.5 KHz PRF and perfused 90 min following ultrasound stimulation to capture peak cFos expression. Control mice with identical devices implanted were not stimulated before perfusion. Mice were first overdosed with isoflurane before undergoing transcardial perfusions with 4 °C phosphate-

buffered saline (PBS) followed by 4% paraformaldehyde (PFA) in PBS. Intact brains were postfixed with the probe still inside for 72 h to facilitate visualization of the probe tract. Tissue sections (50 μm thickness) were then collected using a Leica VT1000 S vibratome equipped with a platinum-coated double-edged blade (Electron Microscopy Sciences, Cat. #72003-01) and set to a maximal speed of 0.9 mm/s. Sections were chosen based on their proximity to the implantation site - where the probe tract could be directly visualized and those directly anterior and posterior to it. Sections were stained for cFos[69] and then mounted onto micro slices (VWR International, LLC). Vectashield HardSet Mounting Medium with DAPI was applied and slides were coverslipped. Slices were stained with primary antibodies 1:1000 rabbit anti-cFos (Synaptic Systems), 1:1000 chicken anti-GFP (Invitrogen), and secondary antibodies 1:200 Alexa 555 anti-rabbit (Invitrogen); 1:200 Alexa 488 anti-chicken (Invitrogen). All slides were given 12 h to dry at room temperature before imaging. Slices were imaged utilizing a confocal microscope (Zeiss LSM800, Germany) at the 20× objective. Cell counts were calculated for the entire dCA1 layer across the medial-lateral axis of the implanted hippocampus. cFos positive cells were quantified using QuPath, a machine-learning-based bioimage analysis pipeline used to classify regions of interest[70]. Histological images were chosen at random to serve as training data in the QuPath program. cFos positive cells were identified and counted by the algorithm with the experimenter blind to treatment and context groups.

## In vivo modulation of nigrostriatal dopamine release and immunohistochemistry

Adult (P42) female mice received intracranial stereotactic AAV injections under isoflurane anesthesia. AAV expressing hSyn-GRAB-DA2m (AAV2/5, Boston Children's Hospital Viral Core) was injected bilaterally into the dorsal striatum (250 nL/site, $1 \times 1013$ vg/mL titer) at the following coordinates: AP + 0.85 mm, ML: ± 1.80 mm, DV from bregma: −3.10 mm, and then implanted with optic fiber cannulas (200 μm core, 0.50 NA) above the injection site at DV −3.00 mm. Three weeks later, the ImPULS probe was slowly lowered into the SNc at the following coordinates: AP −3.10 mm, ML ± 1.00 mm, DV −4.20 to 4.40 mm. Mice were maintained at 1.0% isoflurane anesthesia throughout the recording protocol. Stimulation trials were recorded when the probe reached 200 μm above the SNc (control) and when the probe made contact with the SNc. Pulsed (PRF 1500 Hz, 50% duty factor) stimulation of a set duration in seconds (514 kHz, 10 V) was delivered in each trial using a microcontroller (Teensy 4.0) running a custom script to trigger the output of the function generator and control the pulse repetition frequency, duty factor, trigger signals for the photometry inputs, and stimulation duration. Fiber photometry data were acquired with 410 and 470 nm μLEDs (40 μW power from the tip of the connector) at 90 fps (8.50 ms exposure, gain 1.0) using the RWD tricolor fiber photometry system (R820). Using a 60 s time window in the RWD analysis software, DF/F and Z-score data were computed after smoothing (W = 15), baseline correction (β = 8), and motion correction with the isosbestic signal. Using custom MATLAB scripts, mean Z-score fluorescence and standard error of stimulation-triggered responses were calculated and plotted. Heatmaps were generated using a bin size of 50 ms and a normalized colormap (0–1) based on the maximum and minimum Z-score fluorescence.

At the completion of the recording, mice were deeply anesthetized and perfused transcardially. Specimens were postfixed overnight in 4% PFA and then cryoprotected using a 30% sucrose solution. Serial coronal sections (80 μm thickness) of the DS and SNc were collected by cryostat sectioning. Tissue sections containing the DS were stained with NeuroTrace Nissl 435/455 (1:500, Thermo Fisher, Cat. #N21479) to label cell bodies. Sections containing the SNc were stained for TH and NeuroTrace Nissl. Briefly, sections were incubated with a rabbit anti-TH (1:1500, Millipore, Cat. #AB152) primary antibody, followed by a

donkey anti-rabbit Alexa Fluor 647 (1:500, Thermo Fisher, Cat. #A-31573) secondary antibody and NeuroTrace Nissl 435/455 (1:500, Thermo Fisher, Cat. #N21479) to label neuronal cell bodies. Histological images were acquired using a Zeiss LSM700 confocal microscope.

## Statistics & reproducibility

One-way ANOVA measure was used to quantify the difference between two independent groups with ($n >= 3$ mice): our control condition of implantation without stimulation and our test condition of implantation with a stimulation protocol. Cells were counted across $n = 4$ coronal slices proximal to the implantation site to ensure adequate capture of the stimulation region.

Repeated-measures ANOVA was used to assess the average AUC across stimulation trials for each mouse from baseline against the period of stimulation. Sidak's post-hoc multiple comparisons test was used to compare paired measurements across the two time points. One data point was excluded due to a significant inhibitory dopamine signal caused by stimulation that was not reproduced. These results are described in the Supplementary Information and Discussion.

No statistical method was used to predetermine the sample size. No other data was excluded from the analyses. The experiments were not randomized. The Investigators were not blinded to allocation during experiments and outcome assessment. Representative micrograph images are chosen to illustrate anatomical localizations. Quantifications of image contents are expressed when necessary. All protocols and parameters used in the analysis are included within the manuscript. Supplementary materials include the raw datasets, analysis scripts, and Supplementary Figs. and tables.

## Reporting summary

Further information on research design is available in the Nature Portfolio Reporting Summary linked to this article.

# Data availability

All data supporting the findings of this study are available within the article and its supplementary files but can be found at https://doi.org/10.6084/m9.figshare.25571583. Any additional requests for information can be directed to, and will be fulfilled by, the corresponding authors. Source data are provided with this paper.

# Code availability

All code supporting the findings of this study will be made available upon request to the corresponding authors. Custom code and documentation is provided at https://doi.org/10.5281/zenodo.11094313.

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

## Acknowledgements

This work was supported by MIT Media Lab Consortium funding. J.F.H. was supported by NIH Neurobiological Engineering Training Program Grant #5T32EB019940. K.A.C. was supported by the BCS Alder (1972) Graduate Student Fellowship and the National Science Foundation Graduate Research Fellowship. C.D. thanks to her late aunt, Dogrucan Caliskanoglu, who lost her life due to brain cancer in 2012, for inspiring this work since then. We thank Dr. Hyunsu Park for technical discussions and assistance with piezoelectric microfabrication. The authors would also like to thank Sophia Shen for assistance with microfabrication and Marine Kaufmann for assistance with mouse surgery and perfusion. We would like to thank the lab members of Conformable Decoders: Colin

Marcus, Jin-hoon Kim, Aastha Shah, and David Sadat for scientific discussion and manuscript review, Dr. Fernando Fernandez from Boston University for discussion about Calcium imaging, and Dr. Kenji Shibata from SCIOCS Company Limited, Sumitomo Chemical group, Japan for discussion about KNN. This work was performed in part at the MIT.nano.

## Author contributions

C.D. conceived the research idea, designed research methodology and aims, and directed all research activities. C.D., J.F.H., and M.O.G.N. designed the experiments. H.E.D. provided insights on device design parameters. J.F.H. and M.O.G.N. fabricated the devices, performed the characterization of devices and performed COMSOL multiphysics simulation. E.C-H performed in vitro primary neuron experiments and hydrophone measurements. S.B.O. assisted in adaptive voltage experiments for accelerated aging and fatigue tests. B.W., J.F.H., and M.O.G.N. performed the 2-p Imaging & Slice experiment ex vivo. E.A.R., A.C.-M., J.F.H., M.O.G.N., and M.S. performed the in vivo cFos experiments. K.A.C., J.F.H., M.O.G.N., and A.A.P. performed the in vivo dopamine (fiber photometry) experiment. J.F.H., M.O.G.N., K.A.C., E.A.R., A.C.-M., E.C.-H., and B.W. contributed in data analysis with the input from J.A.W., M.G.S., F.W., S.R., and C.D. All the authors contributed to writing the manuscript. C.D. supervised the overall project.

## Competing interests

The authors declare no competing interests.
