## [Peer Review File · Nature Communications]

An implantable piezoelectric ultrasound stimulator (ImPULS)
for deep brain activationREVIEWER COMMENTS

Reviewer #1 (Remarks to the Author):

Using ultrasound stimulation via wearable devices or implantable devices for deep brain activation is an interesting research topic recently. In this manuscript, the authors proposed an implantable piezoelectric ultrasound stimulator that generates an ultrasonic focal pressure to modulate the activity of neurons. However, there is lack of substantial advancements in the realm of US stimulation technology. The reviewer is unable to endorse the publishing of this manuscript, at least, for the present stage. The specific comments are listed in the following.

1. Materials selection. There seems not a clear novelty from piezoelectric thin film. The KNN as a biocompatible piezoelectric thin film have been investigated as the implantable devices. Authors need to provide references to show the KNN exhibited better performance than doped PZT. Only using the commercial thin film, without any doping or performance improvement methods, the reviewer thought there is no novelty from the material selection.
2. The motivation of using PMUT. It is not clear why a single element of PMUT is selected as the function device. Generally, bulky transducers with the same dimension can achieve better performance, such as tiny bulky transducers for IVUS as a great example. PMUT is usually widely used with a larger number of elements for imaging or stimulation. A single element PMUT definitely showed limited spatial resolution and directionality. Authors need to explain whether the single element with complex microfabrication steps a wise approach is or not.
3. Device design. The size of the stimulation region and working frequency is a crucial component that is essential for the proper functioning of this device. As demonstrated, the stimulated region is located approximately 140 micrometers from the focal point and approximately 200 micrometers laterally, as determined by the technical limitations of the device. Authors did not provide sufficient reason to support these design parameters. Authors did not mention the proper or required working frequency and minimum/maximum power intensity in the introduction. Therefore, it is hard to convince reviewers that the working frequency around 840 kHz is selected in propose or not. In addition, if the physical dimensions of the device cannot change, how this device be suitable for different

requirements of larger regions of the brain tissue?

4. Form factors. One of the novelty in this work is the authors proposed a flexible PMUT as the implantable device. As we know and approved by the experiment, both of the working frequency and the peak intensity of the device decreased a lot from air to water medium.

Changing the supporting substrate from Si wafer to soft SU-8 can also reduce the performance due to lacking the strong support for the vibration. The cable connection with flexible substrate and serpentine electrode is a good design, however, it is nothing related to the form factor of the tip area of the PMUT. The work did not show strong benefit from the flexible PMUT itself. Reviewer suggested to make the same dimension of the Si-based PMUT (rigid in tip area) to show the performance, biocompatibility, stimulation results, as the reference. The disadvantage of the hard probe is critical to prove the necessity of the flexible probe, even compromising the performance.

5. in vivo studies. The experimental design is too simple to show various circumstances, such as the various distance between device and cell, the only three mice without statistical analysis, diverse population of different cells, long-term stability on the cells not only in the PBS, etc.

Reviewer #2 (Remarks to the Author):

In this study, Hou and colleagues introduced an “implantable piezoelectric ultrasound stimulator” (ImPULS) that can be applied to modulate the activity of neurons. They demonstrated ImPULS can activate ex vivo and in vivo mouse hippocampal neurons.

Moreover, they also revealed a time-locked modulation of nigrostriatal dopamine release by stimulating the SNc.

Thank you to the authors for their submitted manuscript, which I enjoyed reading. This is an interesting and timely study by introduces a new way for precise neuromodulation in the deep brain, which is an important and contemporary research question with strong clinical implications. The manuscript is well-written and structured, with solid data analysis. I only have a few comments that are outlined below.

1. As the authors point out in the manuscript, the limits of other implanted deep brain

stimulation techniques is that “the sensitivity of surrounding tissue can decrease significantly over time due to biofouling and corrosion”. Whether ImpULS may cause similar issues has not been investigated. Additional experiments should be carried out to estimate whether the neural response to ImpULS stimulation will be significantly reduced after a period of stimulation (maybe a few days), and whether this effect is reversible after stop administrating the stimulation. This is important to help understand the safety and durability of this novel implanted stimulation approach.

2. During the accelerated aging test, from my understanding, the ImpULS device was not under working. An additional experiment under the same parameter setting but with the device ON may be helpful for further understanding the durability of the ImpULS.

3. As the authors reported “...exposure to 302.4 billion cycles of a sine wave in 7 days 175 results in a 40% lower amplitude of initial displacement...”, just curious, have the authors tested whether there is a way to compensate for displacement loss by increasing the voltage or adjusting other parameters? If so, an adaptive voltage regulation based on the length of time the device is used may further extend the service life of the device and stabilize the stimulation effect of the device.

4. In the “temperature rise” experiment, what is the baseline temperature of the water medium when performing the measurement?

5. Is the error bar in Figure 2g the standard deviation? Better to be included in the figure legend.

6. The captions of Figure 4b and 4c are reversed.

7. I totally understand sometimes it is annoying to obtain opposite findings in the same experiment and I do appreciate the authors reported both the excitatory and inhibitory modulation of nigrostriatal dopamine release. However, in my opinion, additional experiments on maybe an extra two mice will help people better understand the potential rules of the outcomes. At least these results will be helpful for future ultrasound stimulation

research.

Reviewer #4 (Remarks to the Author):

This study reports an implantable piezoelectric ultrasound stimulator for deep brain stimulation. Results showed that this implantable piezoelectric ultrasound stimulator led to activation of neurons in hippocampal slices and anesthetized mice, and modulation of nigrostriatal dopamine release in SNc. The conclusion is that ultrasound stimulator can modulate the neurons of deep brain region. A few key questions remain outstanding, which preclude the publication of this study.

1. What is basis of ultrasound causing activation of neurons, but not astrocytes or microglial cells? Addressing this question is key to validate the procedure and this study.
2. The maximum acoustic pressure is 100 kPa and can activate the neurons, which is inconsistent with the publication (<https://doi.org/10.1002/adbi.201800041>) that demonstrates that the threshold of activation of neurons is 0.25 MPa. This inconsistency has to be resolved.
3. The chosen ultrasound parameters (i.e. 500 kHz, 10-50% duty cycle, 1500 Hz pulse repetition frequency) have not been justified. It should be described in detail that how were the ultrasound parameters chosen in this paper.
4. More evidence should be provided to support the statement that implantable piezoelectric ultrasound stimulator is safe. It would be a good idea to perform cell viability staining of the brain slices to demonstrate the safety of ultrasound stimulation.
5. In Fig.3, Control group or Sham group should be clarified. Fig.3c is three groups, while Fig.3b only provide two group. Please add the results of 500kHz group.
6. The field of ultrasound neuromodulation has recently undergone major skepticism regarding auditory confounds. Since many modalities of sensory stimulation (1.5 kHz PRF within the hearing range of mice) are known to promote dopamine release this should be included in the discussion; I did not see anything related to this present.
7. Sample sizes are too small to be convincing, especially for biological experiments. A minimum sample size of 6 mice per group is recommended.

Contents

Authors' response to Reviewer #1:.....	2
Comment #1-1:.....	2
Comment #1-2.....	3
Comment #1-3:.....	6
Comment #1-4:.....	8
Comment #1-5:.....	9
Comment #1-6:.....	12
Authors' response to Reviewer #2:.....	16
Comment #2-1:.....	16
Comment #2-2.....	20
Comment #2-3:.....	21
Comment #2-4:.....	23
Comment #2-5:.....	23
Comment #2-6:.....	23
Comment #2-7:.....	24
Authors' response to Reviewer #4:.....	26
Comment #4-1:.....	26
Comment #4-2:.....	27
Comment #4-3:.....	29
Comment #4-4:.....	31
Comment #4-5:.....	34
Comment #4-6:.....	35
Comment #4-7:.....	37

Authors' response to Reviewer #1:

Using ultrasound stimulation via wearable devices or implantable devices for deep brain activation is an interesting research topic recently. In this manuscript, the authors proposed an implantable piezoelectric ultrasound stimulator that generates an ultrasonic focal pressure to modulate the activity of neurons. However, there is lack of substantial advancements in the realm of US stimulation technology. The reviewer is unable to endorse the publishing of this manuscript, at least, for the present stage. The specific comments are listed in the following.

Our response: We thank the reviewer for her/his detailed summary and appreciate the valuable comments below. We believe that these comments helped us greatly to strengthen our manuscript, with regards to how we justify our device design decisions. All our responses and modifications to comply with the reviewer's comments are given below with additional data and associated supportive references.

Comment #1-1:

Materials selection. There seems not a clear novelty from piezoelectric thin film. The KNN as a biocompatible piezoelectric thin film have been investigated as the implantable devices. Authors need to provide references to show the KNN exhibited better performance than doped PZT. Only using the commercial thin film, without any doping or performance improvement methods, the reviewer thought there is no novelty from the material selection.

Our response: We thank the reviewer for this important question on material selection. Indeed, we chose KNN over PZT for not only being biocompatible but also having piezoelectric properties that are comparable or greater than that of PZT (ref 1-1, 1-2, 1-3). A comparison of the d_{33} value and Curie temperature of different piezoelectric materials (including PZT) are given below:

Material	d_{33} (pC/N)	T_c ($^{\circ}$ C)
PZT	470–610	300–400
KNN	300–690	350
BaTiO ₃	180-500	130
ZnO	26.7-110	6
AlN	5.5	1150
PVDF-TrFE	20-30	110

From the comparison, it is clear that KNN has comparable piezoelectric performance to the most widely used piezoelectric material, PZT. KNN also possesses superior biocompatibility with regard to the cytotoxicity of its breakdown byproducts compared to PZT (ref 1-4). Furthermore, its high Curie temperature enables advanced fabrication techniques to create device architectures such as the piezoelectric micromachined ultrasound transducer (pMUT), which greatly enhances

the piezoelectric properties of the device without poling or chemical modification. Indeed, piezoelectric devices with KNN have been fabricated, implanted, and evaluated for biocompatibility (ref 1-5). However, this work is the first to characterize a functional implant with the ability to stimulate neural tissue in a designed resonant frequency range. Finally, as part of this revision we further demonstrate the safe actuation of these devices in tissue after a period of 14 days. (Supplementary Figure 23)

In sum, other piezoelectric ceramics and polymers have biocompatible properties, such as BaTiO₃, ZnO, and PVDF, however, they either have inferior electromechanical efficiency and/or thermal processability compared to KNN. These design parameters make KNN the best material choice over PZT, especially for microfabricated implantable neurostimulation devices with the pMUT architecture.

References:

1-1: Tokay O. , Yazıcı M. A review of potassium sodium niobate and bismuth sodium titanate based lead free piezoceramics. *Materials Today Communications*. **31**, 103358, (2022). Supplementary reference 3.

1-2: Yang J., et al. Large piezoelectric properties in KNN-based lead-free single crystals grown by a seed-free solid-state crystal growth method. *Applied Physics Letters* **108.18**, 182904, (2016). Supplementary reference 4.

1-3: Hu C., et al. Ultra-large electric field–induced strain in potassium sodium niobate crystals. *Science advances* **6.13**, eaay5979 (2020). Supplementary reference 5.

1-4: Jeong C. K., et al. Comprehensive biocompatibility of nontoxic and high-output flexible energy harvester using lead-free piezoceramic thin film. *APL Mater.* **5**, 074102 (2017). Supplementary reference 6.

1-5: Chen W., et al. Fabrication of Biocompatible Potassium Sodium Niobate Piezoelectric Ceramic as an Electroactive Implant. *Materials (Basel)*. **10(4)**, 345 (2017). Supplementary reference 7.

Our modification to the manuscript: To address the reviewer’s comment we added a supplementary note (Supplementary Note 1) and five new references in the supplementary information: Supplementary references 3-7.

Comment #1-2.

The motivation for using PMUT. It is not clear why a single element of PMUT is selected as the function device. Generally, bulky transducers with the same dimension can achieve better performance, such as tiny bulky transducers for IVUS or ICE as great examples. PMUT is usually

widely used with a larger number of elements for imaging or stimulation. A single element PMUT definitely showed limited spatial resolution and directionality. Authors need to explain whether the single element with complex microfabrication steps a wise approach is or not.

Our response: We thank the reviewer for this perceptive comment on our choice of a single pMUT transducer. As the reviewer pointed out, the transducers used in IVUS can have good performance in terms of spatial resolution, but please note that even with the smallest conventional piezoelectrics used for IVUS, the resulting probe is still in the near-millimeter scale (ref 1-6). This is due to the electrical wiring and backing layers needed to drive a phased array of pMUTs or miniaturized bulk piezoceramics, respectively. While these devices achieve higher pressures, they are at least an order of magnitude larger than our probe in every dimension and are not designed to target neurons directly adjacent to where they are implanted. Our strategy is desirable for minimizing device cross-sectional area for minimally-invasive implantation while achieving sufficient acoustic pressure and precise targeting.

We do agree that a single-element pMUT precludes the ability to generate a combined focal point with beam steering capabilities, however, our single pMUT structure already provides the desired directionality for therapeutic purposes due to its air-filled backing and higher electromechanical coupling coefficient and energy transfer efficiency to the medium compared to bulk piezoceramics (ref 1-7). Without a piezoelectric actuation structure that includes an in-plane clamped boundary, the resonant frequency of the thin films from thickness-mode vibration alone would be several MHz, which is above the range where acoustic stimulation can occur efficiently. Sub-MHz frequencies from 300 kHz to 1 MHz are widely chosen for stimulating neurons (ref 1-8, 1-9, 1-10, 1-11). However, in a bulk piezoceramic transducer, this thickness mode resonance dependence makes devices millimeters thick, and achieving directionality requires matching and backing layers that further increase the overall transducer dimension significantly. Considering all these requirements, we have chosen a pMUT structure for designing our implantable device.

We also agree that the demonstration of an array of pMUT elements on a flexible substrate may show the versatility of our device and fabrication process when the goal is a large volumetric region of brain stimulation. To comply with the comment, we showed the scalability of our fabrication process with an array of 9 elements. The microscopic image of the fabricated array and resonance profile of array under LDV is shown in supplementary fig. 3.

References:

1-6: Peng C., Wu H., Kim S., Dai X., Jiang X.. Recent Advances in Transducers for Intravascular Ultrasound (IVUS) Imaging. *Sensors*. **21(10)**, 3540, (2021).

1-7: Qiu Y., et al. Piezoelectric Micromachined Ultrasound Transducer (PMUT) Arrays for Integrated Sensing, Actuation and Imaging. *Sensors*. **15(4)**, 8020-8041, (2015). Supplementary reference 12.

1-8: King, R. L., Brown, J. R., Newsome, W. T. & Pauly, K. B. Effective Parameters for Ultrasound-Induced In Vivo Neurostimulation. *Ultrasound Med Biol.* **39**, 312–331 (2013). Reference 15 in the manuscript.

1-9: Tufail Y., et al. Transcranial pulsed ultrasound stimulates intact brain circuits. *Neuron* **66.5**, 681-694, (2010). Reference 22 in the manuscript.

1-10: Blackmore J., Shrivastava S., Sallet J., Butler C. R., Cleveland R. O. Ultrasound neuromodulation: a review of results, mechanisms and safety. *Ultrasound in medicine & biology.* **45.7**, 1509-1536 (2019). Reference 23 in the manuscript.

1-11: Mohammadjavadi M., et al. Elimination of peripheral auditory pathway activation does not affect motor responses from ultrasound neuromodulation. *Brain stimulation.* **12(4)** 901-910, (2019). Reference 24 in the manuscript.

Our modification to the manuscript: To address the reviewer’s comment, we added a new supplementary figure (**supplementary fig. 3**) and added corresponding text **in page 5, lines 104-115** of the manuscript.

Supplementary Figure 3 | a, Microscopic image of microfabricated array with 9 piezoelectric elements. Scale, 200 μm. **b**, Laser doppler vibrometer (LDV) response of 9 elements under a periodic chirp excitation in the 0 - 2 MHz range, showing the displacement of the unique elements

is in phase. **c**, Average frequency response of the array elements, showing resonant frequency close to 500 kHz.

Comment #1-3:

Device design. The size of the stimulation region and working frequency is a crucial component that is essential for the proper functioning of this device. As demonstrated, the stimulated region is located approximately 140 micrometers from the focal point and approximately 200 micrometers laterally, as determined by the technical limitations of the device. Authors did not provide sufficient reason to support these design parameters. Authors did not mention the proper or required working frequency and minimum/maximum power intensity in the introduction. Therefore, it is hard to convince reviewers that the working frequency around 840 kHz is selected in propose or not.

Our response: We thank the reviewer for their comments on the device dimensions, the stimulation area, and the rationale behind frequency selection. This is a worthwhile opportunity for us to clarify how we chose the device dimensions. According to our hydrophone pressure measurements and acoustic pressure simulations, our focal point spans 50 to 100 micrometers laterally and reaches approximately 50 micrometers in the axial direction (Fig. 2d, e). We demonstrate that this focal zone stimulates neurons directly adjacent to the device when implanted (Fig. 3 & 4). The size of this focal zone allows us to target the activation of small groups of neurons significantly enough to activate well-studied distant neural circuits (Fig 4).

It has been reported that the most significant modulation of neurons occurs with the application of ultrasound at frequencies less than 1 MHz (ref 1-12, 1-13). We have added these references to the introduction as well. The driving frequency of 500 kHz in water is well described in the literature as an appropriate frequency for transcranial neuromodulation in mouse neural cells (ref 1-8, 1-9, 1-10, 1-11), as it presents a good tradeoff between skull transmission (more efficient in ranges below 1 MHz) and spatial selectivity (wavelength-dependent focus size decreases with frequency increase). Although our proposed implantable device bypasses the skull and is focused on a volume $<100 \mu\text{m}^3$, we chose to design a device with 500 kHz resonant frequency to reproduce or explore similar established protocols for neuronal activation.

The focal zone of ImpULS was designed to stimulate the volume of several neurons, while optimizing the resonant frequency for neuron sensitivity.

With repeated experiments characterizing devices with different cavity diameters but the same overall device form factor, we demonstrated that we can accurately predict the target resonant frequency while submerged in water (Fig. 2a,b and supplementary fig. 5). As shown in Fig. 2a,b, the resonance frequency shifts by 40.4% when in water versus air. We, therefore, choose 840 kHz as the resonance frequency of our device in air which shifts to 500 kHz in water medium as verified by both simulation and our repeated experiments with impedance analysis and laser

Doppler vibrometry (LDV) measurements. Since the brain tissue has a similar stiffness to 0.6% agar gel, which is a medium slightly more viscous than water, we further measured the resonance of ImpULS in an agar gel medium. We did not notice any significant shift in resonance frequency in gel as compared to the water medium as shown in figure below and therefore assume that any resonance frequency changes between water and brain tissue are insignificant and within the operating bandwidth of the device.

References:

1-8: King, R. L., Brown, J. R., Newsome, W. T. & Pauly, K. B. Effective Parameters for Ultrasound-Induced In Vivo Neurostimulation. *Ultrasound Med Biol.* **39**, 312–331 (2013). Reference 15 in the manuscript.

1-9: Tufail Y., et al. Transcranial pulsed ultrasound stimulates intact brain circuits. *Neuron* **66.5**, 681-694, (2010). **Reference 22 in the manuscript.**

1-10: Blackmore J., Shrivastava S., Sallet J., Butler C. R., Cleveland R. O. Ultrasound neuromodulation: a review of results, mechanisms and safety. *Ultrasound in medicine & biology.* **45.7**, 1509-1536 (2019). **Reference 23 in the manuscript.**

1-11: Mohammadjavadi M., et al. Elimination of peripheral auditory pathway activation does not affect motor responses from ultrasound neuromodulation. *Brain stimulation.* **12(4)** 901-910, (2019). **Reference 24 in the manuscript.**

1-12: Ye, P. P., Brown, J. R. & Pauly, K. B. Frequency Dependence of Ultrasound Neurostimulation in the Mouse Brain. *Ultrasound Med Biol* **42**, 1512–1530 (2016). Reference 16 in the manuscript.

1-13: Tufail, Y., Yoshihiro, A., Pati, S., Li, M. M. & Tyler, W. J. Ultrasonic neuromodulation by brain stimulation with transcranial ultrasound. *Nat Protoc* **6**, 1453–1470 (2011). Reference 39 in the initial submission of the manuscript. **Modified as Reference 21 in the manuscript.**

Our modification to the manuscript: We added a new supplementary fig. (**Supplementary fig. 5**) to show shift in resonance frequency among different mediums (air, water, and 0.6% agar gel) and added new text in the the manuscript: **Page 3, lines 53-56; Page 6, lines 132-133, 138-140; Page 15 lines 341-344.**

We also added three new references in the manuscript. **References 22, 23, 24**

Supplementary Figure 5 | Shift in resonance frequency in different mediums (air, water and 0.6% agarose gel). Almost no shift in frequency is observed in water vs agarose gel medium. N = 4.

Comment #1-4:

In addition, if the physical dimensions of the device cannot change, how this device be suitable for different requirements of larger regions of the brain tissue?

Our response: We thank the reviewer for raising this important question, which centers on the ability of the device strategy to scale to larger brains and regions. As demonstrated in supplementary fig. 3, we are able to make arrays with repeated elements in custom configurations to address larger regions of tissue. With this architecture, we maintain individual addressability to subgroups of the stimulation area. The physical dimension of the device can be changed while keeping its resonance frequency constant at 500 kHz (in water or agarose gel medium) to meet the requirements of stimulating larger regions of the brain tissue

Our modification to the manuscript: To address the reviewer’s comment, we added a new supplementary figure (**Supplementary fig. 3**) and added corresponding text in **page 5 lines 104-115** of the manuscript:

Supplementary Figure 3 | a, Microscopic image of microfabricated array with 9 piezoelectric elements. Scale, 200 μm. **b**, Laser doppler vibrometer (LDV) response of 9 elements under a periodic chirp excitation in the 0 - 2 MHz range, showing the displacement of the unique elements is in phase. **c**, Average frequency response of the array elements, showing resonant frequency close to 500 kHz.

Comment #1-5:

Form factors. One of the novelty in this work is the authors proposed a flexible PMUT as the implantable device. As we know and approved by the experiment, both of the working frequency and the peak intensity of the device decreased a lot from air to water medium. Changing the supporting substrate from Si wafer to soft SU-8 can also reduce the performance due to lacking the strong support for the vibration. The cable connection with flexible substrate and serpentine electrode is a good design, however, it is nothing related to the form factor of the tip area of the PMUT. The work did not show strong benefit from the flexible PMUT itself. Reviewer suggested making a Si-based PMUT (rigid in tip area) with the same area to show the performance, biocompatibility, stimulation results, as the reference. The disadvantage of the hard probe is critical to prove the necessity of the flexible probe, even compromising the performance.

Our response: Thank you to the reviewer for their balanced comments on the device flexibility and material choice. We appreciate that the reviewer has nicely pointed out that quantifying the performance of a rigid device is critical to proving the necessity of a flexible probe. The reviewer is correct in pointing out that it is just the SU-8 encapsulated pMUT alone that is implanted in tissue, and the flexible interconnect serves as a minimal and lightweight connection to the device.

SU-8 is a common material used for biocompatible implanted devices due to its low stiffness (Young's Modulus, 100x less than Si), photo-patternability, and encapsulation ability (ref 1-14, 1-15). These factors are important for implanted devices, as previous studies reported that, due to its flexibility, the mechanical damage caused by SU-8 needles in the rat's brain during its insertion is lower than that caused by rigid Si needles (ref 1-16). In chronic studies, SU-8 devices outperformed Si devices in minimizing tissue damage from mechanical mismatch (ref 1-17). The frequency-dependence of an edge-clamped circular pMUT can be modeled with $f = \frac{\alpha}{2\pi r^2} \sqrt{\frac{D_E}{\rho h}}$

where $D_E = \frac{Eh^3}{12(1-\nu^2)}$, where D_E is the flexural rigidity, α is the resonance mode constant, h is the diaphragm thickness, E is the effective Young's modulus, ν is the Poisson's ratio, ρ is the effective density of the diaphragm. As observed, to maintain a sub-MHz resonance frequency with Si, the diaphragm thickness must be reduced at the expense of insulation or piezo thickness; or the diaphragm radius must be increased significantly, which decreases the spatial resolution of the device. The precedent literature serves as the basis for us to make an informed decision of fabricating flexible pMUT using SU-8 instead of using rigid Si. To comply further with reviewer comments, we performed a COMSOL multiphysics simulation for resonance frequency and acoustic pressure of a Si pMUT and compared it with the results of our flexible pMUT as shown in attached Supplementary Figure 9 figure.

As seen from the simulation, for the same device dimension, the SU-8-based pMUT shows a dominant resonance frequency at 545 kHz whereas Si-based pMUT has resonance at 1350 kHz. It is well established in the literature that robust stimulation in the brain occurs at a sub-MHz frequency range ideally around 500 kHz (ref 1-8, 1-9, 1-10, 1-11) which helped in choosing SU-8 based neurostimulator design. In addition, as seen in fig b, the acoustic pressure exerted by both probes at their resonance frequency is almost similar, yet SU-8-based probes have the obvious advantage of mechanical flexibility as discussed above. Indeed, it is possible to reduce the resonance frequency of a pMUT structure simply by increasing the diameter of the cavity which we have reported in our original submission (supplementary fig. 6), however, to reduce the resonance frequency of Si-based pMUT to ~500 kHz needs significant increase in cavity size thereby overall device size (ref 1-7). Considering all these, we have decided to fabricate flexible SU-8 based pMUT instead of rigid Si-based pMUT.

References:

1-7: Qiu Y., et al. Piezoelectric Micromachined Ultrasound Transducer (PMUT) Arrays for Integrated Sensing, Actuation and Imaging. *Sensors*. **15(4)**, 8020-8041, (2015). Supplementary reference 12.

1-8: King, R. L., Brown, J. R., Newsome, W. T. & Pauly, K. B. Effective Parameters for Ultrasound-Induced In Vivo Neurostimulation. *Ultrasound Med Biol.* **39**, 312–331 (2013). Reference 15 in the manuscript.

1-9: Tufail Y., et al. Transcranial pulsed ultrasound stimulates intact brain circuits. *Neuron* **66.5**, 681-694, (2010). **Reference 22 in the manuscript.**

1-10: Blackmore J., Shrivastava S., Sallet J., Butler C. R., Cleveland R. O. Ultrasound neuromodulation: a review of results, mechanisms and safety. *Ultrasound in medicine & biology.* **45.7**, 1509-1536 (2019). **Reference 23 in the manuscript.**

1-11: Mohammadjavadi M., et al. Elimination of peripheral auditory pathway activation does not affect motor responses from ultrasound neuromodulation. *Brain stimulation.* **12(4)** 901-910, (2019). **Reference 24 in the manuscript.**

1-14: Nemani K. V., Moodie K. L. , Brennick J. B., Su A., Gimi B. In vitro and in vivo evaluation of SU-8 biocompatibility. *Mater Sci Eng C Mater Biol Appl.* **33(7)**, 4453-9, (2013). **Supplementary reference 8.**

1-15: Zhao S., et al. Tracking neural activity from the same cells during the entire adult life of mice. *Nat Neurosci.* **26**, 696–710, (2023). **Supplementary reference 9.**

1-16: Fernández L. J., et al. Study of functional viability of SU-8-based microneedles for neural applications. *J. Micromech. Microeng.* **19(2)**, 025007, (2009). **Supplementary reference 10.**

1-17: Huang S-H., Lin S-P. , Chen J-J. J. In vitro and in vivo characterization of SU-8 flexible neuroprobe: From mechanical properties to electrophysiological recording. *Sensors and Actuators A: Physical.* **216**, 257-265, (2014). **Supplementary reference 11.**

Our modification to the manuscript: To address the reviewer’s comment, we added the simulation results as a new supplementary figure (**supplementary fig. 9**) in the manuscript showing the comparison of Si-based vs SU-8-based pMUT, a supplementary note (**Supplementary Note 2**) and **supplementary references 8-12**. We also added sentences in the main text: **Page 7 line 162-166.**

Supplementary Figure 9 | COMSOL Multiphysics simulation of Si-based and SU-8-based pMUT. a, Displacement vs frequency of Si and SU-8-based pMUT showing their resonance behavior. b, Simulated acoustic pressure profile showing a spherical pressure distribution.

Comment #1-6:

in vivo studies. The experimental design is too simple to show various circumstances, such as the various distance between device and cell, the only three mice without statistical analysis, diverse population of different cells, long-term stability on the cells not only in the PBS, etc.

Our response: We thank the reviewer for this important comment. To demonstrate the capability of ImpULS as a novel neuromodulatory tool, we performed an in vivo dopamine photometry experiment (3 mice) and in vivo modulation of hippocampal CA1 neurons (3 mice) and reported in our initial submission. Following the reviewer's comment, we performed the dopamine photometry experiment on 2 additional mice and added the statistical analysis. The time-locked modulation of nigrostriatal dopamine release in 2 additional mice demonstrates the reproducibility of neurostimulation using our ImpULS. The ability for ImpULS to precisely spatially target and activate the SNc is further confirmed by our use of an internal stimulation control, which demonstrates that stimulating tissue 100-200 μm above the SNc does not alter dopamine release in the dorsal striatum. Please note that the main objective of this project was to develop a new neuromodulatory tool where we capitalize on the ultrasound-sensitivity of neural cells as an implantable modality for neurostimulation. We believe that the reproducibility of the included in vivo experiments and statistical data will convince the reviewer of the ability of ImpULS to robustly and reversibly activate neural cells in an acute manner.

To further investigate whether ImpULS stimulation is safe, effective, and durable in the same animal over time, we conducted an experiment to assess activation from ImpULS after 14 days of implantation in vivo. After recovery from the initial device implantation, we applied our maximum condition of stimulation to isolate the cellular response to the vibrating surface and ultrasound after the acute inflammatory response had subsided. We used immunohistochemistry staining to visualize cell markers involved in inflammation and the foreign body response in 4 mice with ImpULS implanted with stimulation and 3 mice with a ImpULS implanted with no stimulation. We also colocalized a cell nucleus biomarker DAPI with cFos staining to validate that cells remain viable and sensitive to stimulation by the device. As shown in supplementary fig. 21, there is minimal response of activated microglia stained by GFAP near the tip of the device where stimulation occurred, yet there is still robust stimulation as indicated by the cFos/DAPI cell counts.

Furthermore, we conducted a cell viability study to assess the biocompatibility of the constituent materials of the ImpULS device. We extracted cortical tissues from embryonic mice and cultured primary neurons on a fixed ImpULS device for a period of 10 days. Cell densities across 6 plates (3 for control and 3 for ImpULS) were assessed and found to stabilize after a cell medium change on day 3. Neurons on both Control and ImpULS plates developed and differentiated neurites stably. We added representative images and normalized cell counts in supplementary Figure 17.

A detailed chronic study of ImpPULS implanted over the course of months in the SNc is one of our major future extensions of this project and will be reported as a separate publication in future..

Our modification to the manuscript: To address the reviewer’s comment, we modified the main “Fig. 4” with new data from the in vivo dopamine experiment and added the statistical analysis (Fig. 4d). We also added two supplementary figures: i) **Supplementary Fig. 17:** Cell Viability Study with Cultured Neurons and ii) **Supplementary Fig. 21:** Activated cells after chronic implantation and terminal stimulation. The corresponding modification in manuscript texts: **Page 4 line 85; Pages 10-11 lines 240-246; Page 12 line 270, 273, lines 279-285; Page 13 line 290, lines 303-314; Page 14 lines 315-316, 319, 323, 325-326, 332-334; Pages 24-25 lines 556-585, line 588; Pages 28-29 lines 650-683;**

We also added new **Reference 70 in the manuscript:** Weaver I. A. , Yousefzadeh S. A., Tadross M. R. An open-source head-fixation and implant-protection system for mice. *HardwareX*. **13**, e00391 (2023).

Fig. 4 | Stimulation of nigrostriatal dopamine release in anesthetized mice

a, Schematic diagram of the experimental approach to stimulate SNc DA neurons, including post-hoc histological validation of on-target implantation and DA2m sensor expression. Scale bar, 500 μm . b, Averaged DA2m fluorescence responses for control (top right) and SNc (bottom right) stimulation trials. Average heatmaps of fluorescence across trials for control (top left, 3 mice) and SNc (bottom left, 3 mice) stimulation trials. c, Full recording trace of Z-score DA2m fluorescence across stimulation trials, with onset and offset of stimulation (5 s, 1500 Hz, 50% duty factor) indicated by solid and dashed red lines, respectively. d, Area under the curve analysis for 5 seconds pre-stimulation versus 5 seconds during stimulation for average control and SNc stimulation trials (3 mice/group). Repeated-measures 2-way ANOVA with Sidak's multiple comparisons test (** $p < 0.001$, **** $p < 0.0001$).

Supplementary Figure 17 | Cell Viability Study with Cultured Neurons. a, Representative microscopic images of cultured embryonic cortical cells on glass plates without ImPULS present

and glass plates with ImpPULS present over a 10 day period. After day 10, cells were fixed and stained for Anti-MAPS2 to visualize neuron somas and neurites with more contrast, Scale bar, 100 μm . **b**, Normalized cell counts over 10 days.

Supplementary Figure 21 | Activated cells after chronic implantation and terminal stimulation. **a**, Representative fluorescence images of cFos, DAPI, and GFAP in a Control Group with ImpPULS implanted but no stimulation and a Stimulation Group with ImpPULS implanted and a 60 s stimulation after 14 days. Stimulation occurs at the end of the estimated device profile shown in dotted lines, Scale bar, 500 μm . **b**, Cell counts for cFos+ cells normalized to DAPI+ cells in dCA1 following stimulation after a period of 14 days post-implantation (N = 3-4 mice per condition; No-stim vs. 500 kHz, 10% duty factor: unpaired t-test p = 0.0258).

Authors' response to Reviewer #2:

In this study, Hou and colleagues introduced an “implantable piezoelectric ultrasound stimulator” (ImPULS) that can be applied to modulate the activity of neurons. They demonstrated ImPULS can activate *ex vivo* and *in vivo* mouse hippocampal neurons. Moreover, they also revealed a time-locked modulation of nigrostriatal dopamine release by stimulating the SNc.

Thank you to the authors for their submitted manuscript, which I enjoyed reading. This is an interesting and timely study by introduces a new way for precise neuromodulation in the deep brain, which is an important and contemporary research question with strong clinical implications. The manuscript is well-written and structured, with solid data analysis. I only have a few comments that are outlined below.

Our response: We appreciate the reviewer’s dedicated time and expertise in conducting a thorough review of our work. All comments are very valuable and insightful for the improvement of our paper. We have carefully addressed each comment with point-by-point replies as follows.

Comment #2-1:

As the authors point out in the manuscript, the limits of other implanted deep brain stimulation techniques is that “the sensitivity of surrounding tissue can decrease significantly over time due to biofouling and corrosion”. Whether ImPULS may cause similar issues has not been investigated. Additional experiments should be carried out to estimate whether the neural response to ImPULS stimulation will be significantly reduced after a period of stimulation (maybe a few days), and whether this effect is reversible after stop administrating the stimulation. This is important to help understand the safety and durability of this novel implanted stimulation approach.

Our response: We thank the reviewer for their comment about the longevity of the device and the sensitivity of the surrounding tissue. We agree that to demonstrate this new form of implanted stimulation, a chronic study of stimulation reversibility is a vital experiment to conduct. Please note that, for the first time, we envisioned to demonstrate ultrasound as an implantable neuromodulation tool. In this initial device, we focused on systematic device design, optimization, *in vitro* characterization for building a systematic understanding of device’s working principle, *in vitro* durability and fatigue testing, and finally, our device’s ability to stimulate neurons *ex vivo* in brain slice and *in vivo* in two different deep brain regions (hippocampus and SNc). With the DA photometry experiments conducted in the SNc we demonstrated the reversibility of the DA signal to baseline levels after the stimulation is stopped. We believe that these demonstrations may suffice to show our device’s ability to stimulate the deep brain and offer an alternative method for neurostimulation. However, we do agree that an assessment of the safety of ImPULS stimulation is crucial for demonstrating ImPULS as a system that can be used for therapeutic stimulation.

To further investigate whether ImPULS stimulation is safe, effective, and durable in the same animal over time, we conducted an experiment to assess activation from ImPULS after 14 days of implantation *in vivo*. After recovery from the initial device implantation, we applied our maximum

condition of stimulation to isolate the cellular response to the vibrating surface and ultrasound after the acute inflammatory response had subsided. We used immunohistochemistry staining to visualize cell markers involved in inflammation and the foreign body response in 4 mice with a device implanted with stimulation and 3 mice with a device implanted and no stimulation. We also colocalized a cell nucleus biomarker DAPI with cFos staining to validate that cells remain viable and sensitive to stimulation by the device. As shown in the new supplementary fig. 21, there is minimal response of activated microglia stained by GFAP near the tip of the device where stimulation occurred, yet there is still robust stimulation as indicated by the cFos/DAPI cell counts.

Furthermore, we conducted a cell viability study to assess the biocompatibility of the constituent materials of the ImpULS device. We extracted cortical tissues from embryonic mice and cultured primary neurons on a fixed ImpULS device for a period of 10 days. Cell densities across 6 plates (3 for control and 3 for ImpULS) were assessed and found to stabilize after a cell medium change on day 3. Neurons on both Control and ImpULS plates developed and differentiated neurites stably. We added representative images and normalized cell counts in supplementary fig. 17.

In our future studies, we are planning to perform chronic photometry and behavioral experiments to demonstrate the long-term viability of the probe as a neuromodulatory tool and will be published as a follow-up study. In sum, our follow-up experiments assessing the safety of ImpULS through isolating the effects *in vivo* and *in vitro* stimulation meet the initial criteria for reversibility and durability to serve as an alternative to existing implanted stimulation modalities.

Our modification to the manuscript: To address the reviewer's comment we added the following modification in the manuscript: Two new supplementary figures (**Supplementary figs. 17 and 21**), text modifications on **Page 4 line 85; Pages 10-11 lines 240-246; Page 12 line 270, 273, lines 279-285; Page 13 line 290, lines 303-314; Page 14 lines 315-316, 319, 323, 325-326, 332-334; Pages 24-25 lines 556-585, line 588; Pages 28-29 lines 650-683;**

Fig. 4 | Stimulation of nigrostriatal dopamine release in anesthetized mice

a, Schematic diagram of the experimental approach to stimulate SNc DA neurons, including post-hoc histological validation of on-target implantation and DA2m sensor expression. Scale bar, 500 μm . b, Averaged DA2m fluorescence responses for control (top right) and SNc (bottom right) stimulation trials. Average heatmaps of fluorescence across trials for control (top left, 3 mice) and SNc (bottom left, 3 mice) stimulation trials. c, Full recording trace of Z-score DA2m fluorescence across stimulation trials, with onset and offset of stimulation (5 s, 1500 Hz, 50% duty factor) indicated by solid and dashed red lines, respectively. d, Area under the curve analysis for 5 seconds pre-stimulation versus 5 seconds during stimulation for average control and SNc stimulation trials (3 mice/group). Repeated-measures 2-way ANOVA with Sidak's multiple comparisons test (*** $p < 0.001$, **** $p < 0.0001$).

Supplementary Figure 17 | Cell Viability Study with Cultured Neurons. **a**, Representative microscopic images of cultured embryonic cortical cells on glass plates without ImPULS present and glass plates with ImPULS present over a 10 day period. After day 10, cells were fixed and stained for Anti-MAPS2 to visualize neuron somas and neurites with more contrast, Scale bar, 100 μ m. **b**, Normalized cell counts over 10 days.

Supplementary Figure 21 | Activated cells after chronic implantation and terminal stimulation. **a**, Representative fluorescence images of cFos, DAPI, and GFAP in a Control Group with ImPULS implanted but no stimulation and a Stimulation Group with ImPULS implanted and a 60 s stimulation after 14 days. Stimulation occurs at the end of the estimated device profile shown in dotted lines, Scale bar, 500 μm . **b**, Cell counts for cFos+ cells normalized to DAPI+ cells in dCA1 following stimulation after a period of 14 days post-implantation (N = 3-4 mice per condition; No-stim vs. 500 kHz, 10% duty factor: unpaired t-test $p = 0.0258$).

Comment #2-2

During the accelerated aging test, from my understanding, the ImPULS device was not under working. An additional experiment under the same parameter setting but with the device ON may be helpful for further understanding the durability of the ImPULS.

Our response: We thank the reviewer for this very important comment on the aging test. Following the reviewer's comment, we conducted the accelerated aging test again in 75 °C PBS solution but with the device ON for 10 minutes at resonant frequency, 50% duty cycle and 1.5 KHz PRF daily in order to simulate a maximum-stress operating condition in a chronic study. With the introduction of a new stressor (application of continuous signal for 10 minutes), which causes fatigue in piezo elements and added corrosive stress in PBS, we expected a decrease in amplitude similar to the one observed in the fatigue tests (supplementary fig. 12), where we showed that continuous application of sinusoidal signals degrades the device performance. However, with an adaptive voltage strategy (also suggested by the reviewer in his next comment), we can compensate for this loss of piezoelectric performance. As shown in supplementary fig. 12, to maintain the stable displacement of ImPULS, we readjusted the applied voltage to sustain the same device performance measured on day 0 at the driving voltage of 2 V, which

corresponded to a displacement of 120 nm as captured with the Laser Doppler Vibrometer (LDV). We captured microscopic images daily to illustrate the integrity of the device after 7 days of the accelerated aging test. Measurements were taken in 3 devices.

Our modification to the manuscript: To address the reviewer’s comment we added a new supplementary figure (**supplementary fig. 12**) and added the corresponding text modifications in the manuscript: **Pages 9 lines 193-199; Page 22 line 525; Page 23 lines 531-540.**

Supplementary Figure 12 | Aging test with adaptive voltage. a, Microscopic image of ImPULS taken each 24 h apart during aging test with adaptive voltage. Scale bar, 100 μ m. **b,** Required applied voltage to maintain the displacement of ImPULS at 120 nm in 7 days. Error bar represents standard deviation in measurements, N = 3. **c,** Normalized displacement in 7 days. Displacement was normalized by dividing the measured displacement by 120 nm, which was the target displacement to be maintained throughout the experiment. Error bar represents standard deviation in measurements, N = 3.

Comment #2-3:

As the authors reported “...exposure to 302.4 billion cycles of a sine wave in 7 days results in a 40% lower amplitude of initial displacement...”, just curious, have the authors tested whether there is a way to compensate for displacement loss by increasing the voltage or adjusting other

parameters? If so, an adaptive voltage regulation based on the length of time the device is used may further extend the service life of the device and stabilize the stimulation effect of the device.

Our response: We thank the reviewer for their insightful suggestions on the stimulation effect of the device and posing the ability of the device to compensate for piezoelectric degradation with adaptive voltage regulation. As addressed in the previous comment, we measured the displacement of the device using Laser Doppler Vibrometer (LDV) on day 0 of the experiment and established the displacement of 120 nm (achieved at 2 V voltage on day 0) as the target displacement to be reached by readjusting the voltage throughout the days in case of performance loss. We performed the electromechanical fatigue test again keeping the device ON for 5 days, and recorded daily the readjusted voltage needed to achieve the target displacement (equivalent to day 0 performance). As seen in the new supplementary fig. 14, we were able to compensate for the expected displacement loss and could stabilize the performance of our device. Please note that, after 5 days we had to stop the experiment because of the current limitation of LDV equipment where we can apply maximum 10 V from the internal generator of LDV.

As a future study we will develop piezoelectric driver circuits to replace the benchtop electronics with miniaturized electronics to facilitate the chronic behavioral study. In order to demonstrate adaptive voltage regulation in vivo in a future study, we will add a current-sense module calibrated to the results of this study to enable reproducible neurostimulation by maintaining the same ultrasound pressure.

Our modification to the manuscript: To address the reviewer's comment we added a new supplementary figure (**supplementary fig. 14**) and the following modifications in the text of the manuscript: **Page 9 line 204-210; Page 23 lines 542, 546-549.**

Supplementary Figure 14 | Fatigue test of ImpULS with adaptive voltage. a, Required applied voltage to maintain the displacement of ImpULS at 120 nm in 5 days. N = 1. **b,** Normalized

displacement in 5 days. Displacement was normalized by dividing the measured displacement by 120 nm, which was the target displacement to be maintained throughout the experiment.

Comment #2-4:

In the “temperature rise” experiment, what is the baseline temperature of the water medium when performing the measurement?

Our response: We thank the reviewer for this comment to clarify our temperature experiment baseline. The baseline water temperature was measured by a digital thermometer and controlled by a water conditioner that maintains low dissolved gas content, temperature (22.5 °C), deionization, and suspended particulates (Onda Aquas-10). The fiber-optic hydrophone is only able to measure changes in temperature, so the digital thermometer reference and water conditioning system were used to collect a baseline temperature.

Our modification to the manuscript: To address the reviewer’s comment we added the text modification in the manuscript: **Page 9 line 213.**

Comment #2-5:

Is the error bar in Figure 2g the standard deviation? Better to be included in the figure legend.

Our response: We thank the reviewer for this comment and pointing out this discrepancy. We are sorry for this confusion. Yes, the error bar represents the standard deviation. Following the reviewer’s suggestion, we included standard deviation in the figure legend.

Our modification to the manuscript: To address the reviewer’s comment we added the correction in the Figure 2g caption of the manuscript: **Page 36 lines 928-929.**

Comment #2-6:

The captions of Figure 4b and 4c are reversed.

Our response: We thank reviewer for pointing out our unintentional mistake in typing. We corrected the captions of Figure 4b and 4c that were reversed in initial submission. In addition, we also added a new sub figure (Figure 4d) with statistical data after performing new experiments on two extra mice as suggested by the reviewer in his/her next comment (Comment #2-7).

Our modification to the manuscript: To address the reviewer’s comment, we modified the figure 4 caption in the revised manuscript: **Pages 37 lines 935-936.**

Comment #2-7:

I totally understand sometimes it is annoying to obtain opposite findings in the same experiment and I do appreciate the authors reported both the excitatory and inhibitory modulation of nigrostriatal dopamine release. However, in my opinion, additional experiments on maybe an extra two mice will help people better understand the potential rules of the outcomes. At least these results will be helpful for future ultrasound stimulation research.

Our response: We thank the reviewer for this important comment and encouraging words. To demonstrate the capability of ImpULS as a novel neuromodulatory tool, we performed an in vivo dopamine photometry experiment (3 mice) and in vivo modulation of hippocampal CA1 neurons (3 mice) and reported in our initial submission. Following the reviewer's suggestion, we reproduced the dopamine fiber photometry experiment in two additional mice to enable statistical comparisons. The time-locked modulation of nigrostriatal dopamine release in 2 additional mice demonstrates the reproducibility of the excitatory modulation using our ImpULS. The ability for ImpULS to precisely spatially target and activate the SNc is further confirmed by our use of an internal stimulation control, which demonstrates that stimulating tissue 100-200 μm above the SNc does not alter dopamine release in the dorsal striatum.

Our modification to the manuscript: To address the reviewer's comments, we modified the main "Fig. 4" with new data from the in vivo dopamine experiment and added the statistical analysis. The modification in the revised manuscript is as follows: **Page 12 line 270, 273, lines 279-285; Page 13 line 290, lines 303-314; Page 14 lines 315-316, 319, 323, 325-326, 332-334.**

Fig. 4 | Stimulation of nigrostriatal dopamine release in anesthetized mice

a, Schematic diagram of the experimental approach to stimulate SNc DA neurons, including post-hoc histological validation of on-target implantation and DA2m sensor expression. Scale bar, 500 μm . b, Averaged DA2m fluorescence responses for control (top right) and SNc (bottom right) stimulation trials. Average heatmaps of fluorescence across trials for control (top left, 3 mice) and SNc (bottom left, 3 mice) stimulation trials. c, Full recording trace of Z-score DA2m fluorescence across stimulation trials, with onset and offset of stimulation (5 s, 1500 Hz, 50% duty factor) indicated by solid and dashed red lines, respectively. d, Area under the curve analysis for 5 seconds pre-stimulation versus 5 seconds during stimulation for average control and SNc stimulation trials (3 mice/group). Repeated-measures 2-way ANOVA with Sidak's multiple comparisons test (*** $p < 0.001$, **** $p < 0.0001$).

Authors' response to Reviewer #4:

This study reports an implantable piezoelectric ultrasound stimulator for deep brain stimulation. Results showed that this implantable piezoelectric ultrasound stimulator led to activation of neurons in hippocampal slices and anesthetized mice, and modulation of nigrostriatal dopamine release in SNc. The conclusion is that ultrasound stimulator can modulate the neurons of deep brain region. A few key questions remain outstanding, which preclude the publication of this study.

Our response: We thank the reviewer for conducting a comprehensive review of our manuscript and for offering insightful comments. We believe that these remarks have significantly contributed to the improvement of our paper with regard to elucidating the mechanisms underlying ultrasound neurostimulation with our particular device. Below, we have provided detailed responses addressing each comment made by the reviewer, along with corresponding modifications in the manuscript. Additionally, we have incorporated supplementary data and relevant references to further support our work.

Comment #4-1:

What is basis of ultrasound causing activation of neurons, but not astrocytes or microglial cells? Addressing this question is key to validate the procedure and this study.

Our response: We thank the reviewer for their important comment on the types of neural cells activated by ultrasound. The ability of ultrasound waves to activate neural cells of various types has been demonstrated in several past works with explorations into mechanisms encompassing the activation of mechano-sensitive PIEZO and TRP channels, demonstrated in vitro (ref 4-1, 4-2) and in vivo with transcranial focused ultrasound (ref 4-2 , 4-3). These channels exist in both neurons and astrocytes, but studies have shown that different neural cell types might respond differently to US stimulation. Zhu et al. (2023) (ref 4-3) demonstrated that knocking out the highly mechano-sensitive PIEZO channels in neurons resulted in the loss of US modulation sensitivity while knocking out PIEZO channels in astrocytes did not. Lee et al. (2023) (ref 4-4) and Oh et al. (2019) (ref 4-2) demonstrated that TRPA-channels from astrocytes can be ultrasonically activated and are sufficient to indirectly excite neurons via glutamate release. Therefore, we hypothesize that ultrasound can activate both neurons and astrocytes, although likely employing different mechanisms. Indeed, the ultrasound frequency and stimulation parameters can potentially be tuned to achieve a degree of cell selectivity as has been demonstrated by other groups (ref 4-5).

In our study, the calcium indicator GCaMP7F labels primary excitatory neurons, and we demonstrated activation of excitatory cells when the transducer is placed adjacent to the neuron bodies of granule cells in the dentate gyrus (supplementary fig. 18). It is important to note, however, that its expression does not imply the lack of activation of astrocytes/glial cells. We achieved robust stimulation using an ultrasound driving protocol that is known to excite neurons (ref 4-6, 4-7), but the mechanism of action can be partially driven by indirect excitation via astrocytes' gliotransmitters release.

In the scope of this work, we have demonstrated the ability of a new implantable and spatially precise device to cause neuron excitation, and we envision that further studies can help elucidate

the mechanism of action of this activation in the various regions we tested, whether via direct or indirect stimulation.

References:

4-1: Yoo, S., Mittelstein, D. R., Hurt, R. C., Lacroix, J. & Shapiro, M. G. Focused ultrasound excites cortical neurons via mechanosensitive calcium accumulation and ion channel amplification. *Nat Commun* **13**, 493 (2022). Reference 11 in the manuscript.

4-2: Oh, S.-J. et al. Ultrasonic Neuromodulation via Astrocytic TRPA1. *Current Biology* **29**, 3386-3401.e8 (2019). Reference 14 in the manuscript.

4-3: Zhu J., et al. The mechanosensitive ion channel Piezo1 contributes to ultrasound neuromodulation. *Proc Natl Acad Sci USA*. **120(18)**, e2300291120, (2023). **Supplementary reference 13.**

4-4: Lee K., et al. Ultrasonocoverslip: In-vitro platform for high-throughput assay of cell type-specific neuromodulation with ultra-low-intensity ultrasound stimulation. *Brain Stimulation*. **16(5)**, 1533-1548, (2023). **Supplementary reference 14.**

4-5: Murphy K. R., et al. A tool for monitoring cell type-specific focused ultrasound neuromodulation and control of chronic epilepsy. *Proc Natl Acad Sci USA*. **119(46)**, e2206828119, (2022). **Reference 61 in the manuscript**

4-6: Blackmore J., Shrivastava S., Sallet J., Butler C. R., Cleveland R. O. Ultrasound neuromodulation: a review of results, mechanisms and safety. *Ultrasound in medicine & biology*. **45.7**, 1509-1536 (2019). **Reference 23 in the manuscript**

4-7: Manuel T. J. et al. Ultrasound neuromodulation depends on pulse repetition frequency and can modulate inhibitory effects of TTX. *Scientific Reports* **10(1)**, 15347, (2020). **Reference 60 in the manuscript.**

Our modification to the manuscript: To address the reviewer's comment regarding the excitation of neuron cells within the brain tissue, we added the following modification in the "Discussion" section of the manuscript: **Page 15 line 358-362, 364; Page 16 lines 365-366.** We have also added **Supplementary Note 3** to discuss the potential mechanisms of action of ultrasound neuromodulation and **References 60 and 61** in main text and **Supplementary references 13 and 14.**

Comment #4-2:

The maximum acoustic pressure is 100 kPa and can activate the neurons, which is inconsistent with the publication (<https://doi.org/10.1002/adbi.201800041>) that demonstrates that the threshold of activation of neurons is 0.25 MPa. This inconsistency has to be resolved.

Our response: We thank the reviewer for bringing to our attention the need to discuss the activation thresholds and parameters for activating neural cells. We highlight that, within ultrasound driving parameters that are similar to ours, pressures close to 100 kPa have been shown to activate neural circuitry, as exemplified by Tufail et al. (2010) (ref 4-8), who tested pulsed driving frequencies in the 0.25-0.5 MHz range with max pressure of 97 kPa and published a separate protocols paper (reference 21 in the manuscript) (ref 4-9). Notably, they found that their pulsed protocol with PRFs in the 2.5 kHz range was sufficient to activate robust stimulation in hippocampal circuits. In other studies by Yoo et al. (2022) (ref 4-1) in cortical neurons, pressures exceeding 150 kPa were needed to reliably drive neural activation but subthreshold pressures still had modulatory effects.

It is important to note that different regimes in ultrasound driving parameters might create different mechanical effects on the brain tissue. Although the mechanisms of action of ultrasound neuromodulation are not completely understood, it is understood that ultrasound-generated cavitation and acoustic radiation force (ARF) are of main importance in the mechanical pathway for triggering action potentials in neurons through the elastic deformation of cell membranes (ref 4-6). While peak acoustic pressure is an important parameter for those phenomena, the driving frequency and pulse repetition frequency (PRF) are also of critical importance (ref 4-8). The ARF is directly proportional to the driving frequency and the square of the acoustic pressure. The Mechanical Index (index of cavitation likelihood) is directly proportional to the acoustic pressure, and inversely proportional to the square root of the frequency (ref 4-6, 4-10), which is exemplified by the typical operating frequencies of ultrasound neurostimulation under 1MHz.

In the reference that the reviewer graciously provided, Lin et al. (2018) (ref 4-11) investigated the ultrasonic effects on ion channels of cultured pyramidal neurons using a neurostimulation chip that generated standing waves at a 27.38 MHz resonant frequency (standing waves generate more ARF compared to traveling waves, as per Lin et al. 2018). Compared to our 0.1 MPa / 0.5 MHz / traveling waves setup, Lin et al.'s chosen parameters of 0.25 MPa/ 27.38 MHz / standing waves have a much bigger likelihood of being driven purely by ARF effects and not by cavitation or a combination of both. This discrepancy in acoustic wave mode and frequency regime can explain why the activation threshold pressure was different in the distinct setups.

In summary, we believe that the maximum acoustic pressure alone is necessary but not sufficient as an ultrasound parameter to determine the activation threshold of neurons, as other driving parameters such as driving frequency can determine the dominant mechanical effect of ultrasound in the neuronal tissue. As Lin et al. (2018)(ref 4-11) drive the ultrasound in a regime quite different from ours, we believe the inconsistencies in the activation threshold for pressure are reasonable, compared to the setup of our study at much lower frequencies (<1MHz) with precedence for activation in various regions of the brain.

References:

4-1: Yoo, S., Mittelstein, D. R., Hurt, R. C., Lacroix, J. & Shapiro, M. G. Focused ultrasound excites cortical neurons via mechanosensitive calcium accumulation and ion channel amplification. *Nat Commun* **13**, 493 (2022). Reference 11 in the manuscript.

4-6: Blackmore J., Shrivastava S., Sallet J., Butler C. R., Cleveland R. O. Ultrasound neuromodulation: a review of results, mechanisms and safety. *Ultrasound in medicine & biology*. **45.7**, 1509-1536 (2019). **Reference 23 in the manuscript.**

4-8: Tufail Y., et al. Transcranial pulsed ultrasound stimulates intact brain circuits. *Neuron* **66.5**, 681-694, (2010). **Reference 22 in the manuscript.**

4-9: Tufail, Y., Yoshihiro, A., Pati, S., Li, M. M. & Tyler, W. J. Ultrasonic neuromodulation by brain stimulation with transcranial ultrasound. *Nat Protoc* **6**, 1453–1470 (2011). Reference 39 in the manuscript. **Modified as Reference 21 in the manuscript.**

4-10: Radjenovic S. , Dörl G., Gaal M., Beisteiner R. Safety of Clinical Ultrasound Neuromodulation. *Brain Sciences* **12(10)**, 1277, (2022).

4-11: Lin Z., et al. On-chip ultrasound modulation of pyramidal neuronal activity in hippocampal slices. *Advanced Biosystems*. **2(8)**, 1800041, (2018).

Our modification to the manuscript: To address the reviewer’s comment about threshold activation parameters we added the following sentence in the “**Discussion**” section of the manuscript: **Page 15 lines 347-349**. Also, we added a new reference (**Reference 22**)

Comment #4-3:

The chosen ultrasound parameters (i.e. 500 kHz, 10-50% duty cycle, 1500 Hz pulse repetition frequency) have not been justified. It should be described in detail that how were the ultrasound parameters chosen in this paper.

Our response: We thank the reviewer for bringing to our attention that the rationale for the chosen parameters was not clearly described in the text.

The first chosen parameter, the driving frequency of 500 kHz in water, is well described in the literature as an appropriate frequency for transcranial neuromodulation in mice brains (ref 4-6, 4-8, 4-12, 4-13), as it presents a good tradeoff between skull transmission (more efficient in ranges below 1 MHz) and spatial selectivity (focus size decreases with frequency increase). Although our proposed implantable device bypasses the skull and is focused in a volume $<100 \mu\text{m}^3$, we chose to design a device with 500 kHz resonant frequency as it could reproduce or explore similar established protocols for neuron activation. Ye et al. (2016)(ref 4-14) investigated the frequency dependence of ultrasound neuromodulation in the mouse brain, and in the range of 0.3 - 2.9 MHz, demonstrated that the activation success rate was nearly flat at lower frequencies but, at higher frequencies, higher spatial peak intensities were necessary to attain comparable success rates when contrasted with lower ones.

The Pulse Repetition Frequency dependency on ultrasound neuromodulation was explored by Manuel et al. (2020) (ref 4-7), who demonstrated that pulsed stimulation was more efficient than continuous wave stimulation. Among pulsed US, the parameter space that led to the biggest

activation of neurons (quantified by calcium imaging) had PRF of 1500 Hz, center frequency of 500 kHz, acoustic pressure of 100 kPa, and burst duty cycle of 60%, which are very similar to our stimulation parameters.

Taken together, we can see there are a number of parameter investigations for the ultrasonic modulation of neuronal tissue, but no consensus on which optimal parameters should be used. Recommendations for successful parameters of ultrasound neuromodulation encompass a range, and our chosen parameters are contained in the recommendations of both Blackmore et al.'s (2019) (ref 4-6) review and Tufail et al.'s (2011) (ref 4-9) protocol.

Often the choice of parameters is limited by the availability of commercial ultrasound probes or the manufacturing capabilities of research facilities. One advantage of our chosen method of pMUT fabrication is the possibility of creating an ultrasound element or an array of elements in many sizes and center frequencies, a flexibility that will allow for the continuation of parameter investigations in various ultrasound modulation applications. Therefore, for the current work investigating neuron excitation in mice, our ultrasound driving parameters of 500 kHz, 10-50% duty cycle, 1500 Hz pulse repetition frequency are consistent with literature and corroborated our findings of ImpULS's ability to elicit a modulatory response.

References:

4-6: Blackmore J., Shrivastava S., Sallet J., Butler C. R., Cleveland R. O. Ultrasound neuromodulation: a review of results, mechanisms and safety. *Ultrasound in medicine & biology*. **45.7**, 1509-1536 (2019). Reference 23 in the manuscript.

4-7: Manuel T. J. et al. Ultrasound neuromodulation depends on pulse repetition frequency and can modulate inhibitory effects of TTX. *Scientific Reports* **10(1)**, 15347, (2020). Reference 60 in the manuscript.

4-8: Tufail Y., et al. Transcranial pulsed ultrasound stimulates intact brain circuits. *Neuron* **66.5**, 681-694, (2010). Reference 22 in the manuscript.

4-9: Tufail, Y., Yoshihiro, A., Pati, S., Li, M. M. & Tyler, W. J. Ultrasonic neuromodulation by brain stimulation with transcranial ultrasound. *Nat Protoc* **6**, 1453–1470 (2011). Reference 39 in the manuscript. Modified as Reference 21 in the manuscript.

4-12: King, R. L., Brown, J. R., Newsome, W. T. & Pauly, K. B. Effective Parameters for Ultrasound-Induced In Vivo Neurostimulation. *Ultrasound Med Biol*. **39**, 312–331 (2013). Reference 15 in the manuscript.

4-13: Mohammadjavadi M., et al. Elimination of peripheral auditory pathway activation does not affect motor responses from ultrasound neuromodulation. *Brain stimulation*. **12(4)** 901-910, (2019). Reference 24 in the manuscript.

4-14: Ye, P. P., Brown, J. R. & Pauly, K. B. Frequency Dependence of Ultrasound Neurostimulation in the Mouse Brain. *Ultrasound Med Biol* **42**, 1512–1530 (2016). Reference 16 in the manuscript.

Our modification to the manuscript: To address the reviewer's comment, we have added **Supplementary Note 4** to discuss the rationale for our ultrasound parameters choices.

Comment #4-4:

More evidence should be provided to support the statement that implantable piezoelectric ultrasound stimulator is safe. It would be a good idea to perform cell viability staining of the brain slices to demonstrate the safety of ultrasound stimulation.

Our response: We thank the reviewer for the important comment regarding the safety of our implantable device and the suggestion of cell viability experiments.

In the design of our implantable device, we chose to use biocompatible materials that were used in other in vivo studies with mice with no specific adverse effects. Our substrate material, SU-8 is a soft, conformable polymer that has been shown to maintain cell viability when implanted subcutaneously and elicits only a thin fibrous capsule formation with no sustained inflammatory reaction past the implantation-related stress (ref 4-15). For our device, we used SU-8 to encapsulate Au/Cr and Pt electrodes, an approach that was also used by Zhao et al (2023)(ref 4-16), who monitored action potentials in the brain throughout the entire adult life of mice with negligible immune response. In chronic studies, SU-8 devices outperformed silicon devices in minimizing tissue damage from mechanical mismatch (ref 4-17). In addition, our piezoelectric material is KNN, a lead-free ceramic shown to be biocompatible with human fibroblast cells and rat Schwann cells (ref 4-18).

Following the reviewer's suggestion, we additionally conducted two new experiments to assess cell viability, in vitro and in vivo:

To further investigate whether ImpULS stimulation is safe, effective, and durable in the same animal over time, we conducted an experiment to assess activation from ImpULS after 14 days of implantation in vivo. After recovery from the initial device implantation, we applied our maximum condition of stimulation to isolate the cellular response to the vibrating surface and ultrasound after the acute inflammatory response had subsided. We used immunohistochemistry staining to visualize cell markers involved in inflammation and the foreign body response in 4 mice with a device implanted with stimulation and 3 mice with a device implanted and no stimulation. We also colocalized a cell nucleus biomarker DAPI with cFos staining to validate that cells remain viable and sensitive to stimulation by the device. As shown in the figure, there is minimal response of activated microglia stained by GFAP near the tip of the device where stimulation occurred, yet there is still robust stimulation as indicated by the cFos/DAPI cell counts.

Furthermore, we conducted a cell viability study to assess the biocompatibility of the constituent materials of the ImpULS device. We extracted cortical tissues from embryonic mice and cultured

primary neurons on a fixed ImpULS device for a period of 10 days. Cell densities across 6 plates (3 for control and 3 for ImpULS) were assessed and found to stabilize after a cell medium change on day 3. Neurons on both Control and ImpULS plates developed and differentiated neurites stably. We added representative images and normalized cell counts in supplementary fig. 17.

In our future studies, we are planning to perform chronic photometry and behavioral experiments to demonstrate the long-term viability of the probe as a neuromodulatory tool and will be published as a follow-up study. In sum, our follow-up experiments assessing the safety of ImpULS through isolating the effects *in vivo* and *in vitro* stimulation meet the initial criteria for reversibility and durability to serve as an alternative to existing implanted stimulation modalities.

References:

4-15: Nemani K. V., Moodie K. L. , Brennick J. B., Su A., Gimi B. In vitro and in vivo evaluation of SU-8 biocompatibility. *Mater Sci Eng C Mater Biol Appl.* **33(7)**, 4453-9, (2013). Supplementary reference 8.

4-16: Zhao S., et al. Tracking neural activity from the same cells during the entire adult life of mice. *Nat Neurosci.* **26**, 696–710, (2023). Supplementary reference 9.

4-17: Huang S-H., Lin S-P. , Chen J-J. J. In vitro and in vivo characterization of SU-8 flexible neuroprobe: From mechanical properties to electrophysiological recording. *Sensors and Actuators A: Physical.* **216**, 257-265, (2014). Supplementary reference 11.

4-18: Gaukås, N. H. *et al.* In Vitro Biocompatibility of Piezoelectric $K_{0.5}Na_{0.5}NbO_3$ Thin Films on Platinized Silicon Substrates. *ACS Appl Bio Mater* **3**, 8714–8721 (2020). Reference 34 in the manuscript. Modified as Reference 38 in the manuscript.

Our modification to the manuscript: To address the reviewer's comment we added the following modification in the manuscript: Two new supplementary figures (Supplementary figs. 17 and 21), text modifications on Page 4 line 85; Pages 10-11 lines 240-246; Page 12 line 270, 273, lines 279-285; Page 13 line 290, lines 303-314; Page 14 lines 315-316, 319, 323, 325-326, 332-334; Pages 24-25 lines 556-585, line 588; Pages 28-29 lines 650-683;

Supplementary Figure 17 | Cell Viability Study with Cultured Neurons. **a**, Representative microscopic images of cultured embryonic cortical cells on glass plates without ImPULS present and glass plates with ImPULS present over a 10 day period. After day 10, cells were fixed and stained for Anti-MAPS2 to visualize neuron somas and neurites with more contrast, Scale bar, 100 μ m. **b**, Normalized cell counts over 10 days.

Supplementary Figure 21 | Activated cells after chronic implantation and terminal stimulation. **a**, Representative fluorescence images of cFos, DAPI, and GFAP in a Control Group with ImpULS implanted but no stimulation and a Stimulation Group with ImpULS implanted and a 60 s stimulation after 14 days. Stimulation occurs at the end of the estimated device profile shown in dotted lines, Scale bar, 500 μ m. **b**, Cell counts for cFos+ cells normalized to DAPI+ cells in dCA1 following stimulation after a period of 14 days post-implantation (N = 3-4 mice per condition; No-stim vs. 500 kHz, 10% duty factor: unpaired t-test $p = 0.0258$).

Comment #4-5:

In Fig.3, Control group or Sham group should be clarified. Fig.3c is three groups, while Fig.3b only provide two group. Please add the results of 500kHz group.

Our response: We thank the reviewer for this important comment and pointing out this discrepancy in the presentation of the data. For this experiment, in the control group, no ultrasound stimulation is applied as described in “**Stimulation of CA1 in anesthetized mice induces cFos expression**” section (**Page 12 line 268**) in our initial submitted manuscript. We agree that fig. 3b lacked a full stitched image of the region that was congruent with the images shown for the ‘Control’ and ‘500 kHz, 10% duty factor’ group. We have since captured a congruent representative image for the ‘500 kHz’ continuous wave group that can be seen in the revised figure (Fig. 3b (bottom)) and its modified caption.

Our modification to the manuscript: To address the reviewer’s comment we added the following modification in the manuscript: We modified the **fig. 3b** and caption (**Page 36, lines 928-929**).

Fig 3: Robust stimulation of the dCA1 in anesthetized mice.

a, Experimental design and schematic diagram of surgical procedure. **b**, Representative images of hippocampus across experimental conditions: No-stim (top), and 500 kHz, 10% duty factor for 60 s (middle), and 500 kHz, continuous wave for 60s (bottom). Boxed area approximates the dCA1 area used for cell counts. Scale bar, 450 μ m. **c**, cFos+ cells in dCA1 normalized by area across experimental conditions (N = 3-4 mice per condition; No-stim vs. 500 kHz: $p = 0.0506$; No-stim vs. 500 kHz, 10% duty factor: $p = 0.0184$).

Comment #4-6:

The field of ultrasound neuromodulation has recently undergone major skepticism regarding auditory confounds. Since many modalities of sensory stimulation (1.5 kHz PRF within the hearing range of mice) are known to promote dopamine release this should be included in the discussion; I did not see anything related to this present.

Our response: The reviewer's insightful comments regarding auditory confounds are appreciated, and we have revised the **Discussion** section to include our design considerations that address this issue as it relates to ImPULS. Although we attempted to acknowledge the presence of auditory confounds in the introduction, “*Ultrasound, when transmitted from outside the human skull, faces significant scattering and reflection from the skull’s high acoustic impedance¹⁷, which can cause off-target stimulation via conduction through bone and auditory pathways^{18,19} and even traumatic, irreversible brain injury²⁰*”, with references 18 and 19, we agree that further information can be provided about this issue.

While studies involving transcranial ultrasound neuromodulation have demonstrated activation of the auditory cortex in mice even in the absence of application of US in auditory regions (ref 4-19), and also an audible tone that can be reported by human subjects (ref 4-20), there is strong evidence that those confounds are not the ultimate drivers of neuromodulation in our device and that ultrasound can activate neural circuitry without auditory activity. Evidence of the pure ultrasound effects on neurons can be found in the studies of activation of brain slice cultures – like demonstrated in the electrical activity under ultrasound of hippocampal slices in Tyler et al. (2008) (ref 4-21) and also in our present study – besides the evidence that genetically-deafened mice under US stimulation maintained the same motor response as hearing mice (ref 4-13).

The investigation of such auditory confounds in transcranial ultrasound neuromodulation provides convincing evidence that they are related to the broadband spectral components of rapidly varying pulsed waves when they are transmitted through the skull (ref 4-13, 4-20, 4-22). The modulation of the envelope signal, changes in PRF and the topology of the skull are all factors that influence the transmission of sound waves to the ear canal through bone conduction, offering a plausible explanation of how transcranial US modalities have the potential ability to transmit signals to the auditory channels, even if the signal is not intended to be in the audible range. Mohammadjavadi et al. 2019 (ref 4-13) were able to create a mask that effectively eliminates auditory brainstem responses by smoothing the varying US pulse on the ramp up and down.

For our neuromodulation strategy, we circumvent potential auditory confounds due to transcranial transmission by introducing a highly spatially selective and implantable device. With a focal volume of $<100 \mu\text{m}^3$ after which the acoustic pressure is near ambient levels, we do not expect our ultrasound beams to travel to the auditory channel regions to induce auditory responses by avoiding transmission through the skull altogether.

References:

4-13: Mohammadjavadi M., et al. Elimination of peripheral auditory pathway activation does not affect motor responses from ultrasound neuromodulation. *Brain stimulation*. **12(4)** 901-910, (2019). Reference 24 in the manuscript.

4-19: Sato, T., Shapiro, M. G. & Tsao, D. Y. Ultrasonic Neuromodulation Causes Widespread Cortical Activation via an Indirect Auditory Mechanism. *Neuron* **98**, 1031-1041.e5 (2018). Reference 18 in the manuscript.

4-20: Braun V., Blackmore J., Cleveland R. O., Butler C. R. Transcranial ultrasound stimulation in humans is associated with an auditory confound that can be effectively masked. *Brain stimulation*. **13(6)**, 1527-1534, (2020). **Reference 23 in the manuscript.**

4-21: Tyler W. J. et al. Remote excitation of neuronal circuits using low-intensity, low-frequency ultrasound. *PloS One*. **3(10)**, e3511, (2008).

4-22: Hesselink J. et al. Investigating the impact of skull vibrations on motor responses to focused ultrasound neuromodulation in small rodents and methods to mitigate them. *Phys. Med. Biol.* **68**, 135013, (2023).

Our modification to the manuscript: To address the reviewer's comment on auditory confounds we added the following modification to the manuscript: **Page 16 lines 365-379.**

Comment #4-7:

Sample sizes are too small to be convincing, especially for biological experiments. A minimum sample size of 6 mice per group is recommended.

Our response: We thank the reviewer for this important comment on sample size. To demonstrate the capability of ImPULS as a novel neuromodulatory tool, we performed an in vivo dopamine photometry experiment (3 mice) and in vivo modulation of hippocampal CA1 neurons (3 mice) and reported in our initial submission. Following the reviewer's suggestion, we reproduced the dopamine fiber photometry experiment in two additional mice to enable statistical comparisons. The time-locked modulation of nigrostriatal dopamine release in 2 additional mice demonstrates the reproducibility of neurostimulation using our ImPULS. The ability for ImPULS to precisely spatially target and activate the SNc is further confirmed by our use of an internal stimulation control, which demonstrates that stimulating tissue 100-200 μm above the SNc does not alter dopamine release in the dorsal striatum. We believe that the reproducibility of the included in vivo experiments and statistical data will convince the reviewer of the ability of ImPULS to robustly and reversibly activate neural cells in an acute manner.

Our modification to the manuscript: To address the reviewer's comments, we modified the main "Fig. 4" with new data from the in vivo dopamine experiment and added the statistical analysis. The modification in the revised manuscript is as follows: **Page 12 line 270, 273, lines 279-285; Page 13 line 290, lines 303-314; Page 14 lines 315-316, 319, 323, 325-326, 332-334.**

Fig. 4 | Stimulation of nigrostriatal dopamine release in anesthetized mice

a, Schematic diagram of the experimental approach to stimulate SNc DA neurons, including post-hoc histological validation of on-target implantation and DA2m sensor expression. Scale bar, 500 μm . b, Averaged DA2m fluorescence responses for control (top right) and SNc (bottom right) stimulation trials. Average heatmaps of fluorescence across trials for control (top left, 3 mice) and SNc (bottom left, 3 mice) stimulation trials. c, Full recording trace of Z-score DA2m fluorescence across stimulation trials, with onset and offset of stimulation (5 s, 1500 Hz, 50% duty factor) indicated by solid and dashed red lines, respectively. d, Area under the curve analysis for 5 seconds pre-stimulation versus 5 seconds during stimulation for average control and SNc stimulation trials (3 mice/group). Repeated-measures 2-way ANOVA with Sidak's multiple comparisons test (*** $p < 0.001$, **** $p < 0.0001$).

REVIEWER COMMENTS

Reviewer #1 (Remarks to the Author):

After conducting an extensive analysis of the revised manuscript and rebuttal letter, the reviewer has reached the conclusion that substantial enhancements have been made to the manuscript. There are still some questions that need to be addressed.

1. The properties comparison in Note 1 should use the parameters of mechanical, dielectric, and piezoelectric properties related to the piezoelectric thin film, not the bulky materials, such as transverse piezoelectric coefficients, and piezoelectric voltage constant. The d_{33} of thin films usually is much smaller than bulky materials. Please check the parameters carefully.
2. Following the concern about the parameters, the related simulation method and all parameters used in COMSOL and parameters used for Figure 2 d-e, and SI figure 6-9, should be listed and updated. In addition, why the simulation curve in the SI figure 6a is a polyline, not a smooth curve? Please show the equation of the simulation. Why displacement vs voltage is not a linear relationship. It seems three curves showed various curvature in SI figure 6b.
3. In SI Figure 8, the displacement is more than 500 nm. However, the peak is around 225 nm in Figure 2b. Please explain the difference.

Reviewer #2 (Remarks to the Author):

The author addressed my concerns well. I would like to thank the authors for the extensive review of their manuscript.

Reviewer #4 (Remarks to the Author):

The authors' comprehensive answers to the previous questions have resolved certain concerns I had. There are also some questions that need to be answered by the authors for the publication of this work.

1. The authors' device with a focal volume of $< 100 \mu\text{m}^3$ will work on a very tiny area and activate neurons in that area. It is doubtful that such a small range and amount of activation

of neurons in functional brain regions could elicit the macroscopic stimulatory effects of ultrasound neuromodulation. In addition, this method improves the spatial resolution of the ultrasound stimulation volume but with invasiveness. The innovation and significance of this device for ultrasound neuromodulation technology needs to be elucidated in more detail.

2. Figure 3 seems incorrect. The authors mention that the focal volume is $< 100 \mu\text{m}^3$, but the scale of Fig. 3B is obviously much larger than that. The authors should have magnified the stimulated area in order to observe the staining results of the ultrasound stimulation area. The authors claim that the dCA1 region of the hippocampus has the robust stimulation, but apparently the cFOS expression in the DG is stronger after ultrasound stimulation. This result needs to be explained. In addition, a small region of enhanced cFOS expression can be observed in the dashed box of the middle panel in Fig. 3B, but it is not observed in the bottom one. It is apparent that the results of these representative images are inconsistent with the statistical results in Figure 3C. The authors need to confirm and revise this result carefully.

3. The authors performed neuronal activation using the ImPULS device generating 100kPa focal pressure. Will increasing the acoustic pressure induce a larger range or more neuronal activation? Further verification is needed that the neuronal activation is due to the ultrasound stimulation produced by the device, such as mechanosensitive ion channel antagonists.

4. As the authors state, there are many parameters used in ultrasound neuromodulation studies, and there is no consensus on the optimal parameters. The exploration of multiple parameters will improve the generalizability, reproducibility and safety of the study results.

5. The statistical methods used need to be added to the legend of Fig. 3 and other supplementary results. Moreover, the legend of Fig. 4 said that 'Repeated-measures 2-way ANOVA with Sidak's multiple comparisons test' was used in this result. Does the experimental data satisfy the conditions for using the statistical method in both Fig. 3 and Fig. 4?

Authors' response to Reviewer #1:

After conducting an extensive analysis of the revised manuscript and rebuttal letter, the reviewer has reached the conclusion that substantial enhancements have been made to the manuscript. There are still some questions that need to be addressed.

Our response: Thank you for the reviewer's comments and acknowledgment of our revisions to the manuscript. We hope that our following responses will answer any remaining questions the reviewer has.

Comment #1-1: The properties comparison in Note 1 should use the parameters of mechanical, dielectric, and piezoelectric properties related to the piezoelectric thin film, not the bulky materials, such as transverse piezoelectric coefficients, and piezoelectric voltage constant. The d_{33} of thin films usually is much smaller than bulky materials. Please check the parameters carefully.

Our response: We thank the reviewer for this comment underscoring the importance of distinguishing the piezoelectric properties of thin films. Residual stress, the thickness, and the Young's modulus of the substrate can drastically affect measurements of piezoelectric performance, which, as the reviewer pointed out, is especially pronounced in thin films. Therefore, the piezoelectric coefficients of bulk materials were used for relative comparison. Another column has been added to describe the reported performance of thin film topologies. However, with the consideration of the pMUT structure, the d_{33} becomes further enhanced due to the residual stress the structure and surrounding medium exert on the piezoelectric thin film. Related prior publications on the mentioned piezoelectric properties are added in the parenthesis for each related column.

Material	d_{33} (pC/N)		T_c (°C)
	Thin Films	Bulk	
PZT	100-150 (1-1)	390-510 (1-9)	300–400
KNN	74-128 (1-2, 1-3)	300-690 (1-10)	350
BaTiO ₃	100 (1-4)	250-500 (1-11)	130
ZnO	12.7 (1-5)	9.93 (1-12)	6
Doped ZnO (Transition Metal)	128 (1-6)	110 (1-13)	280-500 (1-15)
AlN	6 (1-7)	-	1150
PVDF-TrFE	28 (1-8)	20-30 (1-14)	110

References:

- 1-1: Herdier, R., Jenkins, D., Remiens, D., Dupont, M. and Osmont, D. A silicon cantilever beam structure for the evaluation of d_{31} , d_{33} and e_{31} piezoelectric coefficients of PZT thin films. *Sixteenth IEEE International Symposium on the Applications of Ferroelectrics, Nara, Japan*, 725-727 (2007).
- 1-2: Zang, G.-Z., Yi, Z.-J., Du, J., Wang, Y.-F. Co_2O_3 doped $(\text{Na}_{0.65}\text{K}_{0.35})\text{NbO}_3$ piezoceramics. *Materials Letters*. **64(12)**, 1394-1397 (2010).
- 1-3: Egerton, L. and Dillon, D.M. Piezoelectric and Dielectric Properties of Ceramics in the System Potassium—Sodium Niobate. *Journal of the American Ceramic Society*. **42**, 438-442 (1959).
- 1-4: Acosta, M. et al. BaTiO₃-based piezoelectrics: Fundamentals, current status, and perspectives. *Appl. Phys. Rev.* **4 (4)**, 041305 (2017).
- 1-5: Li, Y. et al. Towards high-performance linear piezoelectrics: Enhancing the piezoelectric response of zinc oxide thin films through epitaxial growth on flexible substrates. *Applied Surface Science*. **556**, 149798 (2021).
- 1-6: Pan, F., Song, C., Liu, X.J., Yang, Y.C., Zeng, F. Ferromagnetism and possible application in spintronics of transition-metal-doped ZnO films. *Materials Science and Engineering: R: Reports*. **62 (1)**, 1-35 (2008)
- 1-7: Anggraini, S.A., Uehara, M., Hirata, K., Yamada, H. and Akiyama, M. Polarity Inversion of Aluminum Nitride Thin Films by using Si and MgSi Dopants. *Sci Rep.* **10**, 4369 (2020).
- 1-8: Hu, X., You, M., Yi, N., Zhang X. and Xiang Y. Enhanced Piezoelectric Coefficient of PVDF-TrFE Films via In Situ Polarization. *Front. Energy Res.* **9** (2021).
- 1-9: Kim, S., and Lee, H. Piezoelectric Ceramics with High d_{33} Constants and Their Application to Film Speakers. *Materials (Basel, Switzerland)*. **14(19)**, 5795 (2021).
- 1-10: Zhang, N., Zheng, T. and Wu, J. Lead-Free (K,Na)NbO₃-Based Materials: Preparation Techniques and Piezoelectricity. *ACS Omega*. **5 (7)**, 3099-3107 (2020).
- 1-11: Dai, B. et al. Piezoelectric grain-size effects of BaTiO₃ ceramics under different sintering atmospheres. *J Mater Sci: Mater Electron*. **28**, 7928–7934 (2017).
- 1-12: Zhao, M.-H., Wang, Z.-L., & Mao, S. X. Piezoelectric characterization of individual zinc oxide nanobelt probed by piezoresponse force microscope. *Nano Letters*, **4(4)**, 587–590 (2004).
- 1-13: Yang, Y. C., Song, C., Wang, X. H., Zeng, F., Pan, F. Giant piezoelectric d_{33} coefficient in ferroelectric vanadium doped ZnO films. *Appl. Phys. Lett.* **92 (1)**, 012907 (2008).

1-14: Zhang, L. et al. Recent Progress on Structure Manipulation of Poly(vinylidene fluoride)-Based Ferroelectric Polymers for Enhanced Piezoelectricity and Applications. *Adv. Funct. Mater.* **33**, 2301302 (2023).

1-15: Straumal, B. B. et al. Ferromagnetic behaviour of ZnO: the role of grain boundaries. *Beilstein journal of nanotechnology.* **7** 1936-1947 (2016)

Our modification to the manuscript: To address the reviewer’s comment, we modified the **Supplementary Note 1: Rationale for choosing KNN as piezoelectric material for ImPULS** and added 15 new references in the supplementary information: **Supplementary references 1-15.**

Comment #1-2: Following the concern about the parameters, the related simulation method and all parameters used in COMSOL and parameters used for Figure 2 d-e, and SI figure 6-9, should be listed and updated. In addition, why the simulation curve in the SI figure 6a is a polyline, not a smooth curve? Please show the equation of the simulation. Why displacement vs voltage is not a linear relationship. It seems three curves showed various curvature in SI figure 6b.

Our response: We thank the reviewer for commenting on the COMSOL simulation parameters. As we mentioned in the original submission, we employed COMSOL Multiphysics (version 6.0) for the simulation of acoustic pressure generated in a water medium by the ImPULS. The model solves the pressure acoustics, electrostatics, and solid mechanics physics for the solution. We added all parameters used for the simulation in tabular form as seen below.

The geometric parameters used in the model:

Layer	Material	Radius (µm)	Thickness (µm)
Passivation layer	SU-8	70	0.5
Top electrode	Au	30	0.25
Piezoelectric layer	KNN	50	1
Bottom electrode	Pt	50	0.25
Membrane layer	SU-8	70	1
Cavity	-	50	20
Backing layer	SU-8	70	20

The properties of materials used in the model:

Properties	SU-8	Au	Pt	KNN

Density(Kg/m ³)	1190	19300	21450	4000
Young's modulus (GPa)	4.02	70	168	65
Poisson's ratio	0.22	0.44	0.38	0.3
Piezoelectric coefficient (C/m ²)				12
Relative permittivity				1500

In the frequency domain study, the model uses the Helmholtz equation to solve for the total acoustic pressure as a part of the COMSOL 6.0 Acoustic Module. We used the physics-controlled extremely fine mesh for the multiphysics simulation.

The reviewer pointed out that the polyline plot of the simulation curve in SI Fig. 6a was unclear. For the COMSOL simulation of the effect of cavity size on resonance frequency, we performed a parametric study where we varied the cavity diameter from 60 to 120 μm with a step size of 10 μm , as seen in the figure below.

After taking points from the COMSOL simulation, a polyline was drawn intersecting maximum points to distinguish the simulation graph from experimental results. To avoid confusion, we removed the polyline and instead plotted cavity simulation results as points and modified the supplementary Fig. 6a in the revised manuscript.

SI Fig. 6b is the experimentally measured displacement of the ImPULS with three different cavity sizes using a LDV. As described in the method section, the devices were mounted on a slab, and water droplets were applied to the device to measure the displacement under hydrostatic equilibrium. Although we have observed a very uniform displacement profile from devices fabricated in the same batch, we should acknowledge that minor batch-to-batch variation from misalignment is still persistent, particularly when the sizes of the cavities of devices vary. The LDV measurement method is also very sensitive to attenuation by surface roughness of the device and attenuation of the reflected laser light. We avoided such errors by increasing the number of devices (N) under measurement and adding the standard deviation, as we did in Fig. 2c. However, for supplementary fig. 6b, our goal was to demonstrate that the ImPULS devices with different cavity sizes have mostly linear displacement with an increased voltage. The small variations that add to the curvature in the line can be attributed to measurement noise. To avoid confusion for the reader, we replaced the line graph by plotting only points and modified the supplementary fig. 6b and caption accordingly.

Our modification to the manuscript: We modified the supplementary figure, **supplementary fig. 6 and caption**, as follows:

Supplementary Figure 6 | Effect of cavity size on resonance frequency and displacement of device **a**, Resonance frequency for varied cavity diameters comparing simulated and experimental results. **b**, Measured displacement with LDV of devices with varied cavity size and applied voltages.

We also added a new supplementary note (**Supplementary Note 5: COMSOL simulation parameters**) in the revised manuscript.

Comment #1-3: In SI Figure 8, the displacement is more than 500 nm. However, the peak is around 225 nm in Figure 2b. Please explain the difference.

Our response: We thank the reviewer for pointing out the varied scales of displacement. Different voltages were applied in these figures, and the range of peak displacement can vary approximately from 100 to 1000 nm (peak to peak) in a voltage range of 2-10V, as demonstrated in Figure 2C.

Figure 2B illustrates how the resonance frequency shifts from an air medium to a water medium when driven by both periodic chirp and a sinusoidal signal, while supplementary fig. 8 illustrates the displacement of the membrane at the resonant frequency (sinusoidal input), highlighting the membrane deformation. However, we acknowledge that we should have mentioned the excitation voltage in the respective figure captions to avoid any confusion. For Figure 2B a 500 kHz sinusoidal signal, 4 V (p-p) was applied while in case of SI Figure 8, 10 V (p-p) was applied.

Our modification to the manuscript: To comply with the reviewer's comment, we added the applied voltage in the mentioned **Figure 2C (Page 36, Line 929)** and **Supplementary Figure 8** captions, respectively:

Fig. 2 | Characterization of implantable piezoelectric ultrasound stimulator (ImPULS).

a, The impedance and phase angle spectra of ImPULS at air and water medium, showing the resonance frequency in both mediums. **b**, Displacement of ImPULS at air and water medium measured using laser doppler vibrometer (LDV) at 4 V (p-p) when the inputs are a periodic chirp (bottom) and a sinusoidal signal (top). **c**, Displacement of ImPULS as a function of input voltage (p-p) with inset showing two-dimensional (2-D) point scan of displacement indicating the lateral resolution of the beam of the device. Error bar represents standard deviation in measurement, $N = 3$. **d**, Simulated acoustic pressure profile of ImPULS showing a spherical pressure distribution. **e**, Comparison of simulated and experimentally measured pressure using a fiber optic hydrophone at different distances. Error bar represents standard deviation in measurement, $N = 3$. **f**, 2-D mapping of pressure generated by ImPULS measured at $z = 15 \mu\text{m}$. Scale bar, $25 \mu\text{m}$. **g**, Microscopic image of ImPULS taken each 24 h apart during aging test (left), and normalized displacement of ImPULS before start of test and after 7 days. Scale bar, $100 \mu\text{m}$. Error bar represents standard deviation in measurement, $N = 3$. **h**, Temperature change in water medium when a continuous sinusoidal signal of 500 kHz at 20 V (p-p) applied to ImPULS. Ultrasound was 'off' for 10 min, 'on' for 10 min and 'off' for 10 min. **i**, 2-D mapping of temperature generated by ImPULS measured at $z = 15 \mu\text{m}$. Scale bar, $25 \mu\text{m}$.

Supplementary Figure 8 | Three representative stages of membrane vibration upon application of sinusoidal signal at 10V (p-p) at fundamental resonance frequency.

Authors' response to Reviewer #2:

The author addressed my concerns well. I would like to thank the authors for the extensive review of their manuscript.

Our response: We would like to thank the reviewer for their time and insightful comments that greatly improved the quality of the manuscript.

Authors' response to Reviewer #4:

The authors' comprehensive answers to the previous questions have resolved certain concerns I had. There are also some questions that need to be answered by the authors for the publication of this work.

Our response: We thank the reviewer for their insightful comments that allowed us to strengthen the manuscript. We hope that our following responses will elucidate any outstanding questions the reviewer has.

Comment #4-1: The authors' device with a focal volume of $< 100 \mu\text{m}^3$ will work on a very tiny area and activate neurons in that area. It is doubtful that such a small range and amount of activation of neurons in functional brain regions could elicit the macroscopic stimulatory effects of ultrasound neuromodulation. In addition, this method improves the spatial resolution of the ultrasound stimulation volume but with invasiveness. The innovation and significance of this device for ultrasound neuromodulation technology needs to be elucidated in more detail.

Our response: We thank the reviewer for the comment about the relation between the microscopic volume of stimulation and the ability to achieve therapeutic macroscopic stimulatory effects. We hope that our explanation and corresponding experiments will convince the reviewer that the potency of ImpULS stimulation is perhaps one of the core strengths of the device.

Our results in the CA1 region of the hippocampus indicate that the focal stimulation elicited broader macroscopic activation effects throughout CA2, CA3, and the dentate gyrus (DG) due to the highly interconnected downstream and back-propagating canonical circuits between the regions. This result highlights the potency of deep brain stimulation with ImpULS in recurrently connected circuits. Considering recurrent connection is a general feature of the mammalian brain, a focal stimulation akin to ImpULS could be conceivable to have a macroscopic effect.

Furthermore, we showed (Fig. 4, supplementary fig. 23) that stimulating a small focal volume in the substantia nigra pars compacta (SNc) with ImpULS elicited robust, time-locked release of dopamine in the dorsal striatum (DS), measured optically with a fluorescent dopamine sensor. Even though DS is a relatively large area, we did not need to target a specific location with the imaging optical fiber. This is because the axon from a single dopaminergic cell, on average, branches nearly 300,000 times and innervates approximately 3% of the total volume of striatal tissue (ref 4-1). As most mammalian neurons in the brain have elaborated axons projecting to multiple downstream targets, it is again conceivable that a microscopic stimulation can have macroscopic effect. It is notable that ImpULS stimulation in the control region, located approximately 150 micrometers above the SNc, failed to elicit time-locked DA release (Fig. 4, supplementary fig. 22). Therefore, at least in the areas of tissue inferior to the device, stimulation does not reach beyond 100 micrometers. We have now added these discussions to the paper.

We also thank the reviewer for recognizing that “this method improves the spatial resolution” to “a focal volume of $< 100 \mu\text{m}^3$ ”, which is indeed one of our key goals for developing ImPULS. The reviewer rightly pointed out that our method is “invasive” compared to non-invasive focused ultrasound modulation technology. However, this approach is less invasive than currently available forms of deep-brain stimulation. Electrical stimulation (e.g., DBS) of deep brain regions requires bulkier probes in the mm-scale with rigid form factors. Furthermore, the stimulation current is proportional to the geometric surface area of the electrodes, which means that neuromodulatory effects potent enough for therapeutic effects in deep brain circuits arise from electrodes in the mm^2 scale and must operate below the physical limits from electrical tissue damage (ref 4-2). Therefore, we believe that this work has justified implanted ultrasound stimulation for the deep brain by drastically reducing invasiveness through size and material-selection without sacrificing potency.

In addition, it is worth recalling here that non-invasive ultrasound neuromodulation through the skull can cause off-target stimulation via conduction through bone and auditory pathways (References 18, 19 in Main). Bypassing these unintended effects is also one of the key factors for designing our implantable ultrasound neurostimulator, ImPULS. We discussed this aspect in the introduction of the manuscript.

References:

4-1: Matsuda, W. et al. Single Nigrostriatal Dopaminergic Neurons Form Widely Spread and Highly Dense Axonal Arborizations in the Neostriatum. *Journal of Neuroscience*. **29 (2)**, 444-453 (2009).

4-2: Cogan, S. F., Ludwig, K. A., Welle, C. G., and Takmakov, P. Tissue damage thresholds during therapeutic electrical stimulation. *Journal of neural engineering*. **13 (2)**, 021001 (2016).

Our modification to the manuscript: To address the reviewer’s comment, we made text modifications (**Page 13, Line 302-303; 305-306; Page 16, Line 376-378**) in the “**Discussion**” section of the manuscript:

Comment #4-2: Figure 3 seems incorrect. The authors mention that the focal volume is $< 100 \mu\text{m}^3$, but the scale of Fig. 3B is obviously much larger than that. The authors should have magnified the stimulated area in order to observe the staining results of the ultrasound stimulation area. The authors claim that the dCA1 region of the hippocampus has the robust stimulation, but apparently the cFOS expression in the DG is stronger after ultrasound stimulation. This result needs to be explained. In addition, a small region of enhanced cFOS expression can be observed in the dashed box of the middle panel in Fig. 3B, but it is not observed in the bottom one. It is apparent that the results of these representative images are inconsistent with the statistical results in Figure 3C. The authors need to confirm and revise this result carefully.

Our response:

We thank the reviewer for their feedback and the opportunity to further explain the results of the ImPULS stimulation in the hippocampus.

We have added new representative images to the main figure, Figure 3, showing a zoomed section of the dCA1 region in the slice adjacent to the implanted probe that shows the local effect of stimulation as well as a better representative image to replace Figure 3b, middle panel, from the segmented slices used for counting. A modification to supplementary figure 20 now shows isolated channels for each of our stimulation conditions as well as close-up images of the groups of cells proximal to the ImPULS transducer. We believe that these images more accurately represent the focal stimulation that ImPULS achieves in dCA1. In addition, regarding the statistical results of Figure 3C, we would like to highlight that the selected region proximal to the implant region in the dCA1 was used for counts across a number of slices. Therefore, regardless of the chosen representative image in Figure 3B, the counts were quantified from a volume directly around the implant probe tract outside known microcircuit pathways.

With regard to the apparent activity in the dentate gyrus (DG), we have added a portion of the following discussion to the manuscript. Contained within the intrinsic circuitry of the hippocampal formation are several parallel processing and feedback networks mediating excitation, inhibition, and disinhibition (ref 4-3). The canonical trisynaptic circuit is comprised of three prominent glutamatergic synapses that connect layer II entorhinal (EC) projections to dentate granule cells via the perforant pathway, dentate (DG) projections to CA3 pyramidal neurons via mossy fibers, and CA3 projections to CA1 pyramidal neurons via Schaffer collaterals. CA1 projections then loop back to the deep layers of the entorhinal cortex or subiculum (ref 4-3). Further complicating the interconnectivity of the hippocampus are projections connecting the subiculum back to CA1, the EC to CA1 and subiculum (the temporoammonic pathway), and local recurrent collaterals within CA3 (ref 4-4). Moreover, an abundance of cell types activated in these signaling pathways express mechanoreceptive channels like Piezo1 (ref 4-5). Previous research has shown that inducing hypersynchrony in CA1 using optogenetic stimulation was sufficient to activate the entire hippocampal formation (ref 4-6). c-Fos results in this experiment were consistent with our work showing increased c-Fos expression in CA1, the DG, and CA3. This suggests our stimulation protocol may have been evoking hypersynchrony and runaway excitation throughout the intrinsic circuitry of the hippocampus detailed above, and serves to explain the high levels of c-Fos expression observed in the DG. In light of these considerations, we decided to restrict our analysis to the CA1 region of the hippocampus such that c-FOS counts were only performed in the CA1 layer most proximal to the implantation site of the imPULS device.

References:

4-3: Knierim, J. J. The hippocampus. *Current Biology*, 25 (23), R1116-R1121 (2005). Reference 60 in the manuscript.

4-4: Xu, X., Sun Y., Holmes T. C. and López, A. J. Noncanonical connections between the subiculum and hippocampal CA1. *J Comp Neurol*. **524(17)**, 3666–3673 (2016).

4-5: Falleroni, F. et al. Mechanotransduction in hippocampal neurons operates under localized low picoNewton forces. *iScience*. **25(2)**, 103807 (2022).

4-6: Osawa, S-I. et al. Optogenetically Induced Seizure and the Longitudinal Hippocampal Network Dynamics. *PLOS ONE*, **8(4)**, e60928 (2013). Reference 61 in the manuscript.

Our modification to the manuscript: To address the reviewer's comment, we updated "**Fig. 3**" (Page 37 Line 944-948) and **Supplementary Figure 20**", and modified the text in the "**Discussion**" (Page 15-16, Line 359-365) section of the manuscript and added two new references (**References 60 & 61**) (Page 35, Line 888-890).

Fig. 3 | Robust stimulation of the dCA1 in anesthetized mice.

a, Experimental design and schematic diagram of surgical procedure. **b**, Representative images of hippocampus across experimental conditions: No-stim (top), and 500 kHz, 500 kHz, 500 kHz, 10% duty factor (middle and bottom).

continuous wave for 60s (**middle**), and 10% duty factor for 60 s (**bottom**). **Yellow dashed** boxed area approximates the dCA1 area used for cell counts. Scale bar, 450 μm . **Color-coded dashed box indicates area shown in magnified section.** Scale bar, 100 μm c, cFos+ cells in dCA1 normalized by area across experimental conditions (**One-way ANOVA**; N = 3-4 mice per condition; No-stim vs. 500 kHz: p = 0.0506; No-stim vs. 500 kHz, 10% duty factor: p = 0.0184).

Comment #4-3: The authors performed neuronal activation using the ImPULS device generating 100kPa focal pressure. Will increasing the acoustic pressure induce a larger range or more neuronal activation? Further verification is needed that the neuronal activation is due to the ultrasound stimulation produced by the device, such as mechanosensitive ion channel antagonists.

Our response:

We appreciate the reviewer's suggestion on acoustic pressure variation, which entails one of the future investigations for our work. ImPULS is a new device platform that operates optimally within several design parameters that enable it to safely elicit macroscopic effects with minimal power consumption, tissue displacement, and temperature change. In fact, understanding how to operate the device at this lower threshold of activation allows the ImPULS platform to maintain power and thermal efficiency.

Our current device is not able to produce pressures higher than 100 kPa due to the polarization saturation limits of the piezoceramic thin film. However, we agree that in the future it would be interesting to build devices that can reach higher pressures and evaluate their performance. We have added a note about this in the Discussion. Regarding the mechanisms of ultrasound neuromodulation, this is a broad topic that has been addressed in numerous previous studies including one led by one of the co-authors of this manuscript (ref 11 in the main text), which have established the ultrasonic origins of the stimulation.

Pharmacological dissection of mechanisms is beyond the scope of this device-focused study, but we have added citations to "Supplementary Note 3: Potential mechanisms of action of Ultrasound Neuromodulation" to guide readers to relevant mechanistic literature (Ref 11, Ref 37, Ref 4-7). We agree that mechanosensitive ion channel antagonists could be used to probe effects in certain cells and cell-environments. However, genetic or pharmacological studies that disable mechanosensitive ion channels simultaneously affect other physiological processes that maintain cell or organism viability (Ref 4-8). With the development of new sonogenetic tools that enhance mechanosensitivity without disabling channel activity (Ref 4-7) and that maintain a diverse environment of mechanosensitive cells (Ref 4-9), we hope to study how the neuronal activation is produced by the ImPULS device in a future work.

References:

4-7: Cadoni, S. et al. Ectopic expression of a mechanosensitive channel confers spatiotemporal resolution to ultrasound stimulations of neurons for visual restoration. *Nat. Nanotechnol.* **18**, 667–676 (2023). Supplementary reference 28

4-8: Dubin, A.E., Schmidt, M., Mathur, J., Petrus, M.J., Xiao, B., Coste, B., and Patapoutian, A. Inflammatory signals enhance piezo2-mediated mechanosensitive currents. *Cell Rep.* **2**, 511–517 (2012). Supplementary reference 29

4-9: Newman, M. et al. Ultrasound Modulates Calcium Activity in Cultured Neurons, Glial Cells, Endothelial Cells and Pericytes. *Ultrasound in Medicine & Biology.* **50(3)**, 341-351 (2024). Supplementary reference 30

Our modification to the manuscript: To address the reviewer’s comment, we made text modifications in the “**Discussion**” section (**Page 17 Line 391-395**) and in the “**Supplementary Note 3: Potential mechanisms of action of Ultrasound Neuromodulation**”.

Comment #4-4: As the authors state, there are many parameters used in ultrasound neuromodulation studies, and there is no consensus on the optimal parameters. The exploration of multiple parameters will improve the generalizability, reproducibility and safety of the study results.

Our response: We thank the reviewer for their relevant comment on optimal parameter choice. Our study is focused on the introduction of a new modality of ultrasound stimulation (flexible, micron-sized implantable devices in deep brain tissue) compared to transcranial-focused ultrasound. Therefore, it was important to maintain parameters that were already reproduced in prior literature (as discussed in Supplementary Note 4) to demonstrate the potency of our new device as a neuromodulatory tool. For a thorough parameter space investigation, a considerable number of mice would be needed to sweep through the different driving parameters (a non-exhaustive list would include center frequency, stimulation intensity and pressure, pulse length and repetition frequency, duty cycle, total stimulation time, etc.). The parameter space investigation is valuable to elucidate the mechanisms of action of ultrasound neuromodulation. However given the time required to complete a full set of parameter titration, we plan to conduct this vital direction as a scope of our future work.

Nevertheless, our device was able to stimulate different areas of the deep brain following the stimulation protocols available in the literature, as demonstrated by our hippocampal neuronal stimulation in *ex vivo* slices and *in vivo* and by stimulation modulating dopamine release in the substantia nigra pars compacta region *in vivo* as well. Regarding safety, we can refer to ITRUSST [Ref 4-10] as an example of an international collaboration of academic and industry experts currently working on establishing standards for the application of safe ultrasonic neuromodulation in humans. ITRUSST published a consensus [Ref 4-11] in which it established the ultrasound parameters that it considers to be biophysically safe, according to an interpretation of existing recommendations for diagnostic ultrasound and implantable devices.

The report determines that devices with a Mechanical Index (MI) less or equal to 1.9 and whose temperature rise due to ultrasound is less or equal to 2° Celsius are to be considered safe. Our

device has an MI of approximately 0.14 (as calculated by $MI = \frac{P_n}{\sqrt{f_c}}$, where P_n is the Peak

Negative Pressure in MPa and f_c is the center frequency in MHz, Ref 4-12), and the maximum temperature change after continuous application of 20 V signal to our ImPULS device is of 0.95°C, as illustrated in Supplementary Figure 15. Therefore, even for the most conservative estimates regarding ultrasound driving parameters, our chosen parameters should be considered safe.

While we agree with the reviewer about the importance of investigating the parameter space for ultrasound neuromodulation, we believe this is a lengthy endeavor that is beyond the scope of this paper, and will be a part of the future work direction. Here, our goal is to provide proof-of-principle for implantable, deep-brain ultrasound stimulation of neurons that is safe and successfully modulates the deep brain right after surgical implantation, as well as after chronic implantation (10 days).

References:

4-10: International Transcranial Ultrasonic Stimulation Safety and Standards, 2021. <https://itrusst.com/aboutus.html> (accessed March 26th, 2024).

4-11: Aubry et al. ITRUSST Consensus on Biophysical Safety for Transcranial Ultrasonic Stimulation. Preprint at <https://arxiv.org/abs/2311.05359> (2023).

4-12: U.S. Food and Drug Administration, “Marketing Clearance of Diagnostic Ultrasound Systems and Transducers”, February 21, 2023. <https://www.fda.gov/regulatory-information/search-fda-guidance-documents/marketing-clearance-diagnostic-ultrasound-systems-and-transducers> (accessed March 26th, 2024).

Our modification to the manuscript: To address the reviewer’s comment, we made text modifications (**Page 17, Line 401-404**) in the “**Discussion**” section of the manuscript:

Comment #4-5: The statistical methods used need to be added to the legend of Fig. 3 and other supplementary results. Moreover, the legend of Fig. 4 said that ‘Repeated-measures 2-way ANOVA with Sidak’s multiple comparisons test’ was used in this result. Does the experimental data satisfy the conditions for using the statistical method in both Fig. 3 and Fig. 4?

Our response: Thank you to the reviewer for pointing out the missing description of our statistical tests for Fig 3. We would like to take this opportunity to explain the reasoning for our statistical analysis and add the appropriate corresponding details to the manuscript.

In Fig. 3, we represented the activity of a region using average co-localized counts of DAPI and cFOS in a segmented region representing dCA1 across multiple coronal slices (n=4). In order to quantify the effect of stimulation of our device apart from the effects of acute implantation, we used a one-way ANOVA measure to quantify the difference between two independent groups with (n>=3): our control condition of implantation without stimulation and our test condition of implantation with a stimulation protocol. The statistical test was mentioned in Line 273 of the manuscript, in the “Stimulation of CA1 in anesthetized mice induces cFos expression” sub-section, and we will also include it in the caption of Fig. 3.

We used repeated measures ANOVA in Fig. 4 because we used the average AUC across stimulation trials for each mouse from baseline (5 seconds before stimulation) vs. the 5 second period of stimulation. This was for two independent variables: stimulation while the probe was approximately 150 micrometers above the SNC (control) vs. stimulation trials once we lowered the probe into the SNC. In other words, these are paired measurements across two independent variables. In order to compare differences both within and between each independent group (across the two time points), we used Sidak's post-hoc multiple comparisons test. An alternative to this post-hoc test method is the Bonferonni, however the Sidak method has more power in that it provides a higher threshold for significance. Given that there are not more than two time points, we believe a paired t-test would be sufficient as well.

Our modification to the manuscript: To address the reviewer's comment, we made caption modifications (**Page 37 Line 944-948**) in Main “**Fig. 3**” :

Fig. 3 | Robust stimulation of the dCA1 in anesthetized mice.

a, Experimental design and schematic diagram of surgical procedure. **b**, Representative images of hippocampus across experimental conditions: No-stim (top), and 500 kHz, **500 kHz, continuous wave for 60s (middle), and 10% duty factor for 60 s (bottom)**. Yellow dashed boxed area approximates the dCA1 area used for cell counts. Scale bar, 450 μm . **Color-coded dashed box indicates area shown in magnified section. Scale bar, 100 μm** **c**, cFos+ cells in dCA1 normalized by area across experimental conditions (**One-way ANOVA**; N = 3-4 mice per condition; No-stim vs. 500 kHz: p = 0.0506; No-stim vs. 500 kHz, 10% duty factor: p = 0.0184).

REVIEWERS' COMMENTS

Reviewer #1 (Remarks to the Author):

I'm ok with the authors' response and the revised manuscript.

The manuscript can be accepted as is.

Reviewer #4 (Remarks to the Author):

The authors have emphasized and answered all my questions.